

# Retrieval of Temperature From a Multiple Channel Pure Rotational Raman-Scatter Lidar Using an Optimal Estimation Method

Shayamila Mahagammulla Gamage[1], Robert. J. Sica[1,2,*], Giovanni Martucci[2], and Alexander Haefele[2,1]

[1]Department of Physics and Astronomy, The University of Western Ontario, London, N6A 3K7, Canada
[2]Federal Office of Meteorology and Climatology, MeteoSwiss, CH-1530 Payerne, Switzerland

**Correspondence:** R. J. Sica (sica@uwo.ca)

**Abstract.**

We present a new method for retrieving temperature from Pure Rotational Raman (PRR) lidar measurements. Our Optimal Estimation Method (OEM) used in this study uses the full physics of PRR scattering and does not require any assumption of the form for a calibration function nor does it require fitting of calibration factors over a large range of temperatures. The only calibration required is the estimation of the ratio of the lidar constants of the two PRR channels (coupling constant) that can be evaluated at a single or multiple height bins using a simple analytic expression. The uncertainty budget of our OEM retrieval includes both statistical and systematic uncertainties, including the uncertainty in the determination of the coupling constant on the temperature. We show that the error due to calibration can be reduced significantly using our method, in particular in the upper troposphere when calibration is only possible over a limited temperature range. Some other advantages of our OEM over the traditional Raman lidar temperature retrieval algorithm include not requiring correction or gluing to the raw lidar measurements, providing a cutoff height for the temperature retrievals that specifies the height to which the retrieved profile is independent of the *a priori* temperature profile, and the retrieval's vertical resolution as a function of height. The new method is tested on PRR temperature measurements from the MeteoSwiss Raman Lidar for Meteorological Observations system in different sky conditions, compared to temperature calculated using the traditional PRR calibration formulas, and validated with coincident radiosonde temperature measurements in clear and cloudy conditions during both day and night time.

## 1 Introduction

High time and space resolution measurements of atmospheric temperature are necessary to improve our understanding of many atmospheric processes, both dynamical and chemical. Radiosounding is the most widely used method for temperature profiling in the troposphere and lower stratosphere, and has the advantage of operation in most weather conditions, but is typically limited to 2 flights per day. Pure rotational Raman (PRR) lidars have excellent vertical and temporal resolution, and can be combined with vibrational Raman channels to determine relative humidity (Mahagammulla Gamage et al.). Lidar temperature





measurements can be assimilated with atmospheric models to improve weather forecasts, as recently demonstrated by Adam et al. (2016).

The traditional Raman lidar temperature retrieval method, introduced by Cooney (1972), uses the ratio of two PRR signals from the Stokes branch which have been corrected for saturation, background and other instrumental effects as required. The

PRR spectrum contains two branches: Stokes and anti-Stokes. Both branches have approximately the same intensity and they are positioned symmetrically in wavelength on either side of the excitation line. The traditional Raman lidar temperature retrieval algorithm requires the assumption of an analytic form of a lidar calibration function whose coefficents are usually determined with external measurements, such as radiosondes (Behrendt, 2005). The calibration function is an approximation of the relationship of the signal ratio and temperature and depends on two or more coefficients. Calibration errors exceeding

0.5 K can arise if the calibration data do not cover a sufficient temperature range (Behrendt, 2005).

Of primary importance is in the calibration of the lidar returns to allow absolute temperature measurements. In the traditional method, the ratio of the corrected photo-counts is fit to a set of corresponding temperature data points usually obtained from radiosondes. The application of the Optimal Estimation Method (OEM) for temperature retrievals using Pure Rotational Raman (PRR) lidar measurements has several advantages over the traditional method, including the ability to find temperature without

assuming an analytic form of the temperature/count ratio relation. Our OEM retrieval does not use the ratio of the counts. Rather we use a forward model which includes complete physics to describe the raw count profiles. For calibration the ratio of the lidar constants, here from referred to as coupling constant, needs to be determined. The coupling constant can in principle be estimated at a single point, such as a nearby flux tower or surface measurement. Our OEM retrieval has other important benefits over the traditional method, as it can directly retrieve ancillary parameters to temperature, such as geometrical overlap, particle

extinction, and the lidar constant. Our OEM is also capable of providing a full uncertainty budget, including both random and systematic uncertainties on a profile-by-profile basis, including the systematic uncertainty introduced in the retrieved temperature by the estimation of the coupling constant. The OEM is an inverse method, and is a standard tool in the retrieval of geophysical parameters from passive atmospheric remote sensing instruments. Recent studies including (Povey et al., 2014; Sica and Haefele, 2015, 2016; Farhani et al., 2019) have shown that OEM can be used to retrieve atmospheric aerosol, water

vapor mixing ratio, middle and upper atmospheric temperature and ozone using lidar measurements.

In Section 2 a brief description of the instrument and the measurements used in this study is presented. Section 3 presents the development of the PRR lidar equation.The development of a forward model for application of the OEM to PRR temperature retrieval is given in Section 4. The OEM-retrieved temperature results from the PRR measurements for different atmospheric conditions are shown in Section 5. A discussion of these results is presented in Section 6 followed by conclusions.

## 30   2   The Raman Lidar for Meteorological Observations

PRR measurements from the RAman Lidar for Meteorological Observations (RALMO), located in Payerne, Switzerland ($46°48'$ N, $6°56'$ E) are used for the OEM temperature profiling. RALMO is a fully automated lidar built at the École Polytechnique Fédérale de Lausanne and operated by MeteoSwiss (Dinoev et al., 2013). It is dedicated to operational meteorology,





validating models and satellite measurements, and climate studies. RALMO has been operating nearly continuously since 2008, with an average data availability of 50%. Data gaps are due to rain and low clouds (approximately 30% of the time), maintenance (1 - 2 days per month) and other occasional technical problems. RALMO consists of a frequency tripled, Q-switched Nd:YAG laser of 354.7 nm producing up to 400 mJ emission energy at 30 Hz repetition rate. The pulse duration is 8 ns. The

laser is operated at 300 mJ energy per pulse to extend the lifetime of the flash lamps from 20 to approximately 60 million shots. The RALMO receiver uses four parabolic mirrors each with 1 m focal length and 30 cm diameter, and it is fiber coupled to a two stage grating polychromator. The data acquisition system consists of photo-multipliers and analog/digital transient recorders from Licel. The system obtains a measurement by adding together 1800 laser shots (every minute) at a vertical resolution of 3.75 m. For a detailed description of the lidar and a detailed validation of the temperature measurements the reader is referred

to Dinoev et al. (2013) and Martucci et al., in preparation.

The returns of the Raman-shifted backscatter arising from rotational energy state transitions of nitrogen and oxygen molecules due to the excitation at the laser wavelength at 354.7 nm are detected in analog and photon counting mode. The high quantum number channel (JH) of RALMO is assigned to the backscattered signals from the energy exchange that occurs in the high quantum states for both the Stokes (355.77-356.37 nm) and anti-Stokes (353.07-353.67 nm) branches. The low quantum

number channel (JL) is assigned to the signals from the energy exchange occurring in the low quantum states in the Stokes (355.17-355.76 nm) and anti-Stokes (353.67-354.25 nm) branches.

## 3   The PRR lidar equation

The backscattered PRR signal is given by the Raman lidar equation:

$$N_{RR,t}(z) = \frac{C_{RR}}{z^2} O(z) n(z) \Gamma_{atm}^2(z) \left( \sum_{i=O_2, N_2} \sum_{J_i} \eta_i \left( \frac{d\sigma}{d\Omega} \right)_\pi^i (J_i) \right) + B_{RR}(z) \tag{1}$$

where the true backscattered PRR signal $N_{RR,t}$, is a function of height $z$, $C_{RR}$ is the lidar constant, $n(z)$ is the number density of the air molecules, $O(z)$ is the geometrical overlap, $\eta_i$ is the volume mixing ratio of nitrogen and oxygen, $\Gamma_{atm}(z)$ is the atmospheric transmission, $\left( \frac{d\sigma}{d\Omega} \right)_\pi^i (J_i)$ is the attenuated differential backscatter cross section for each RR line $J_i$ and $B_{RR}(z)$ is the background of the measured signal. For different lidar systems the background can either be a constant or vary with height. Since air below 80 km is a constant mixture of oxygen and nitrogen, $\eta_i$ is a constant. The lidar constant $C_{RR}$ depends

on the number of transmitted photons, detector efficiency, and the area of the telescope.

The attenuated differential backscatter cross section for Stokes and anti-Stokes line pairs of equal quantum number of the PRR spectrum is expressed as (Penney et al., 1974):

$$\left( \frac{d\sigma}{d\Omega} \right)_\pi^i (J) = \frac{112}{15} \cdot \frac{\pi^4 g_i(J) h c B_{0,i} (v_o + \Delta v_i(J))^4 \zeta_i^2}{(2I_i + 1)^2 kT}$$

$$\times (X^+(J)\tau^+(J_i) + X^-(J)\tau^-(J_i)) \exp \left( \frac{-E_{rot,i}(J)}{kT} \right) \tag{2}$$




where for the Stokes branch

$$X^+(J) = \frac{(J+1)(J+2)}{2J+3} \quad \text{for} \quad J = 0,1,2,\ldots \tag{3}$$

and for the anti-Stokes branch

$$X^-(J) = \frac{J(J-1)}{2J-1} \quad \text{for} \quad J = 2,3,4,\ldots \quad \text{and} \quad X^-(J) = 0 \quad \text{for} \quad J = 0,1. \tag{4}$$

5    $\tau^+(J_i)$ and $\tau^-(J_i)$ are the transmissions of the receiver for the Stokes and anti-Stokes lines $J_i$, respectively. $g_i(J)$ is the statistical weighting factor, which depends on the nuclear spin $I_i$ for each atmospheric constituent, $h$ is Planck's constant, $c$ is the velocity of light, $k$ is Boltzmann's constant, $B_{0,i}$ is the ground-state rotational constant, $v_0$ is the frequency of the incident light, and $\zeta_i$ is the anisotropy of the molecular polarizability. The rotational energy $E_{rot,i}(J)$ for each Stokes and anti-Stokes branch is estimated based on the assumption of a homonuclear diatomic molecule in the quantum state $J$ for nitrogen and 10   oxygen molecules with no electronic momentum coupled to the scattering (Behrendt, 2005).

     The response of photomultiplier tubes used as detectors in lidar systems and operated in the digital mode can become nonlinear at high count rates. In the case of RALMO, the true and observed counts are related by the equation for non-paralyzed systems where $\gamma$ is the counting system dead time:

$$N_{observed} = \frac{N_{true}}{1 + N_{true}\gamma}. \tag{5}$$

15   The dead time of the counting system is the time taken after measuring a photon, before the detector is able to accurately measure another incident photon.

## 4   Application of the OEM for PRR temperature retrieval

### 4.1   Brief review of the optimal estimation method

     The OEM is an inverse method that uses the measurements $y$ to estimate the state (retrieval) variables $x$ of a system via a for- 20   ward model. The forward model $F$ contains all the atmospheric and instrumental physics that describe the measurements. The forward model can include model parameters $b$, which are assumed and not retrieved, and their effect on the retrieved quantity uncertainties can be calculated.

     The measurements are related to the forward model by:

$$y = F(x,b) + \epsilon \tag{6}$$

25   where $\epsilon$ represents measurement noise. Under the assumption that all parameters have Gaussian probability density functions Bayes' theorem can be applied to determine the cost function,

$$cost = [y - F(x,b)]^T S_y^{-1} [y - F(x,b)] + [x - x_a]^T S_a^{-1} [x - x_a], \tag{7}$$





where $S_y$ is the measurement covariance, which describes the random measurement uncertainty and $S_a$ is the *a priori* covariance. The cost function measures the goodness of fit for a solution, and for good models the cost is on the order of unity. For non-linear forward models, the Marquardt-Levenberg method can be used iteratively to minimize the cost of the retrieval (see Section 5.7 in Rodgers (2000) for details).

## 5  4.2  The forward model for a PRR lidar

The forward model describes the measurement as a function of both the state of the atmosphere and instrumental parameters.

The core of our forward model is the lidar equation as presented in Section 3. It is called 4 times to generate the measurements corresponding to high and low quantum numbers, i.e. JH and JL, with digital and analog detection. Analog detection is assumed to be linear and hence the saturation equation (Equation 5) is not applied.

The pressure, $P(z)$, and temperature, $T(z)$, can be taken from either a radiosonde measurement or an atmospheric model. The background noise, $B_{RR}$, is in general a function of height, $z$, but is constant with height for RALMO. Unlike all the other existing forward models for lidar except Povey et al. (2012) (which was designed specifically to determine overlap) we retrieve $O(z)$ the geometrical overlap function in addition to temperature.

The atmospheric transmission, $\Gamma_{atm}(z)$ in Eq.( 1), includes both molecular and particle scattering.

$$\Gamma_{atm}(z) = \exp\left[\int_0^z (\alpha_{mol} + \alpha_{par})dz\right] \tag{8}$$

where $\alpha_{mol}$ is the molecular extinction coefficient and $\alpha_{par}$ is the particle extinction coefficient. The molecular extinction can be expressed using the Rayleigh cross section $\sigma_{Ray}$ and air density $n_{air}$ as:

$$\alpha_{mol} = \sigma_{Ray}.n_{air} \tag{9}$$

where $\sigma_{Ray}$ is calculated using the expressions given by Nicolet (1984).

For each channel the subscript $RR$ is replaced by JL and JH ,the high and low quantum number PRR channels. Then $C_{RR}$, $B_{RR}$ and $J_i$ then have different values.

RALMO detects multiple Stokes and anti-Stokes lines from both nitrogen and oxygen PRR spectrum. Therefore, to determine the attenuated backscatter cross-section in the forward model we require knowledge of the exact number of quantum states detected by each the RALMO PRR channel. From the JH and JL channel characteristics we can calculate the range

of frequency shifts for each channel relative to the elastic wavelength 354.7 nm. Then using the equations given by Herzberg (2013) we can determine the quantum numbers $J_i$ for both nitrogen and oxygen molecules. This calculation process is repeated to determine the $J_i$ numbers for the JL channel of the RALMO. The summary of the findings are given in Table 1.



**Table 1. Return PRR wavelengths detected by the RALMO and the respective quantum lines from nitrogen and oxygen PRR spectrums.**

| Channel | | Nitrogen | Oxygen |
|---|---|---|---|
| JL | Detected wavelengths | 355.17-355.76 nm | 353.67-354.25 nm |
| | Quantum lines (Stokes and anti-Stokes) | 3,4,5,7,8,9 | 5,7,9,11,13 |
| JH | Detected wavelengths | 355.77-356.37 nm | 353.07-353.67 nm |
| | Quantum lines (Stokes and anti-Stokes) | 10,11,12,13,14,15 | 15,17,19,21 |

In order to establish absolute calibration, we define the coupling constant $R$ as the ratio of the lidar constants $C_{JL}$ and $C_{JH}$

$$R = \frac{C_{JH}}{C_{JL}} \tag{10}$$

and use the substitution $C_{JH} = RC_{JL}$. The coupling constant is height-independent and can be determined with no assump-

tions at, if desired, a single altitude using the following equation derived from Eq.( 1).

$$R = \left( \frac{N_{t,JH} - B_{JH}}{N_{t,JL} - B_{JL}} \right) \bigg/ \left( \frac{\sigma_{JH}(T,z)}{\sigma_{JL}(T,z)} \right). \tag{11}$$

The differential cross section terms are determined by applying temperature from a coincident reference temperature, typically from a radiosonde. For a well designed lidar system the coupling constant should be stable over weeks. Unlike the fitting of an analytic calibration function to the data as in the traditional method, $R$ can be estimated at a specific height or range of heights,

which can be over a narrow range of temperature without introducing extrapolation errors. We extensively tested this assertion using both synthetic and real measurements. The results show that the estimation of $R$ is indeed height independent. The value of $R$ is only affected by the measurement noise. Hence, we recommend using a range of heights or a specific height where the photocounts have a high signal-to-noise ratio.

Using $R$ in the forward model allows us to retrieve only one lidar constant, while constraining the two channels to vary so

as to satisfy Eq.( 10). We will see in the next section that any variations or uncertainty in the determination of $R$ introduces an uncertainty on the order of 0.2 K to the retrieved temperature profile.

The retrieval parameters (Table 2) in our OEM algorithm are temperature, background signals (including photo multiplier shot noise, sky background, and offset for analog channels), the lidar constants, dead times of the digital photon counting systems, geometrical overlap, and particle extinction as a function of height. In OEM we can retrieve parameters on a height

grid where the resolution can be same or different than the vertical resolution of the height grid that the measurements obtained. If the retrieval grid is coarser than the measurement grip we use linear interpolation on retrieved quantities when they are required in the forward model.





**Table 2. Values and associated uncertainties for the OEM retrieval and forward model parameters.**

| Parameter | Value | Standard Deviation |
|---|---|---|
| **Measurements** | | |
| Digital | Measured | Poisson Statistics |
| Analog | Measured | Auto Covariance Method |
| **Retrieval Parameters ( *a priori*)** | | |
| Temperature | US Standard Model | 35 K |
| geometrical overlap Function | Estimated using the forward model and measurements | 50% below transition height |
| | | $10^{-3}$ above transition height |
| Particle Extinction | Estimated using measurements | $10^{-6} \mathrm{km}^{-1}$ below transition height |
| | | 50% above transition height |
| Lidar Constants (analog/digital) | Estimated using the forward model | 100% |
| Digital Background Noise | Mean above 50 km | Standard Deviation above 50 km |
| Analog Background Noise | Mean above 50 km | nighttime- Standard Deviation above 50 km |
| | | daytime- normalized standard deviation above 50 km |
| Dead Time | Empirical fitting | 10% |
| **Forward Model Parameters** | | |
| Pressure | Radiosonde | 30 Pa |
| Coupling Constants (analog/digital) | Estimated measurements and sonde temperature | Standard deviation of the coupling constants over a height range |
| Air density | Radiosonde | 1% |

## 4.3 Implementation of the RR temperature retrieval

The OEM solver in the Qpack software package is used for our retrieval (Eriksson et al., 2005). The solver requires the following as inputs: the measurements from each lidar channel and their covariances, *a priori* values for the retrieval parameters and their covariances, model (**b**) parameters and their covariances, and the Jacobians of the forward model.

The lidar measurements from the digital channels follow Poisson counting statistics in the range where the counts are linear. Thus, the measurement variance $S_y$ is equal to the number of photons in each height bin, assuming no correlation between the height bins (that is, the off-diagonal terms in the $S_y$ matrix are zero). The auto-correlation function method of Lenschow et al. (2000) is used to estimate the measurement covariance of the analog and digital measurements in the non-linear region. For both analog and digital channels, the *a priori* backgrounds and their variances are taken as the mean and the variance of the measurements above 50 km height.

The U. S. Standard Atmosphere (NASA, 1976) model temperature profile is normalized to the surface temperature from the coincident sonde temperature, and then used as the *a priori* temperature profile in our retrievals. Due to the high variability of the temperature, a standard deviation of 35 K for all heights is used to specify the covariance matrix for *a priori*. Other choices of *a priori* temperature profile are possible, but as an operational, fully automated lidar system RALMO retrievals must be performed automatically every 30 min, so the choice of the US Standard Model with this covariance simplifies this procedure.

As discussed in Eriksson et al. (2005), the elements of the retrieval and model parameters are often correlated, and some of the covariance matrices should have off-diagonal elements. Off-diagonal elements are parametrized with the correlation length




and an appropriate analytical function describing the decay of the correlation. For this study, we used a tent function with a 1 km correlation length for temperature retrievals (Eriksson et al., 2005).

Molecular and particle extinction terms occur in the atmospheric transmission term of Eq.( 8). An *a priori* particle extinction profile is estimated based on the following expression:

$$\alpha_{par} = LR \cdot \beta_{par} = LR \cdot \beta_{mol} \cdot (\Re_{\beta} - 1) \tag{12}$$

where $LR$ is the lidar ratio, $\beta_{par}$ is the particle backscatter coefficient. $\beta_{par}$ can be related to the backscatter ratio $\Re_{\beta}$ as Whiteman (2003):

$$\Re_{\beta} = \frac{(\beta_{mol} + \beta_{par})}{\beta_{mol}} \tag{13}$$

The backscatter ratio $\Re_{\beta}$ is estimated using the RALMO PRR and elastic measurements. Bucholtz (1995) gives a method for calculating $\beta_{mol}$ using pressure, temperature and Rayleigh cross sections. The Rayleigh extinction cross sections required for $\beta_{mol}$ estimation are computed using the formula of Nicolet (1984). Calculated Rayleigh extinction cross sections are also used to estimate the air density profile used as a $b$ parameter in the forward model, assuming an uncertainty of 1% for the standard deviation.

The lidar ratio $LR$ is chosen based on the $\Re_{\beta}$ values for the given measurements. Typically $\Re_{\beta}$ values inside clouds are greater than 2. Thus, for this study the height at which $\Re_{\beta}$ is first equal to 2 is considered as the height of the cloud base or the height of an aerosol layer (cloud/aerosol layer base height). The cloud/aerosol layer base height is later used to determine the transition height that constrains the range of the geometrical overlap and the particle extinction retrievals. In clear sky conditions (that is if $\Re_{\beta}$ does not exceed 2), $LR$ is assumed to be 80 sr inside the boundary layer and 50 sr elsewhere. In cloudy conditions, $LR$ is assumed to be 20 sr within the clouds present below 6 km. If the cloud is above 6 km, $LR$ is assumed to be 15 sr within the cloud. These choices for lidar ratios are taken from Ansmann et al. (1992a) and Pappalardo et al. (2004). Accurate $LR$ is not crucial, as it is used to estimate an *a priori* particle extinction profile. However, we can calculate a $LR$ profile using the OEM retrieved $\alpha_{par}$ and compare it with the initially chosen $LR$ values to evaluate how good a choice of the initial value is.

The effect of geometrical overlap and particle extinction on the signals are strongly linearly dependent and hence retrieving both parameters simultaneously with the given data channels is not possible unless at least one of the effects is highly constrained. In this work we assume that particle extinction is well known from the backscatter ratio outside clouds and that overlap is well known above the height of full overlap, i.e. above 6 km. Hence we define a transition height at 6 km or cloud base height, whatever is lower, below which particle extinction is retrieved and above which overlap is retrieved. The a priori overlap function is estimated from the measurements in clear sky conditions with little effect due to particles. A 50% standard deviation is used for geometrical overlap below the transition height and a constant value of $10^{-3}$ is used above this height, constraining the geometrical overlap to the a priori above the transition height. For particle extinction, a small standard deviation of $10^{-6}$ km$^{-1}$ is used below the transition height, but a 50% standard deviation is used above this height, allowing the OEM to retrieve exclusively the particle extinction. The *a priori* covariance matrices for both particle extinction and geometrical overlap are determined using a tent function with a 100 m correlation length.





*A priori* lidar constants for the JL analog and JL digital channels are estimated by fitting the measurements generated using the sonde temperature and pressure used in the forward model to the PRR measurements. For analog measurements, the fitting range is between 1.5 to 2 km height. For digital measurements with clear conditions or cloud/aerosol presence above 8 km, 6 to 8 km is used as the fitting range, to insure the photocounts are linear. If the digital measurements contain a cloud or aerosols

in the geometrical overlap region, a fitting range below this is used, typically 3.5 - 4 km height. The fitting uncertainty for each analog and digital lidar constants is used as the variance of the *a priori* lidar constant.

The *a priori* dead times for the two digital photon counting systems are considered to be 3.8 ns, consistent with estimations from previous studies for RALMO and with values specified by the manufacturer (Sica and Haefele, 2016, 2015; Dinoev et al., 2010). The uncertainty in the dead time is taken as 10%. Coincident radiosonde pressure profile are used assuming a 10%

standard deviation. The coupling constants for analog ($R_a$) and digital ($R$) channels are estimated by fitting the ratio of PRR measurements with the ratio of the differential cross section (Eq.( 11)). The coupling constants are estimated using the same fitting range as the lidar constants. Table 2 gives a summary of the parameters and associated uncertainties of the retrieval and **b** parameters used in the forward model.

## 5   Results from the temperature retrieval

We present four different measurement situations which demonstrate the robust nature of our OEM temperature retrieval. Details of each case study are given in Table 3. The RALMO measurements used in the retrievals are added in time over 30 min and to 15 m in height. Analog measurements are used from the surface to 6 km height, while digital measurements are used from from 4 to 28 km. The retrieval grid has a vertical resolution of 60 m at all heights. For all the cases given in Table 3 we used radiosonde measurements that coincide within 1 hour of the lidar measurements for comparison purposes, to estimate

the required *a priori* information, and to determine the forward model **b** parameters (Table 2).

**Table 3. Details of the 4 cases in different sky conditions we present to demonstrate the flexibility of our OEM temperature retrieval.**

| Case | Date | Time (UT) | Sonde Launch (UT) | Sky Condition |
|------|------|-----------|-------------------|---------------|
| 1 | 20110909 | 2300-2330 | 2200 | Clear-nighttime |
| 2 | 20110910 | 1100-1130 | 1100 | Clear-daytime |
| 3 | 20110705 | 2300-2330 | 2300 | Cirrus Cloud ($\sim$ 6 km)-nighttime |
| 4 | 20110621 | 2300-2330 | 2300 | Lower Cloud ( 4 km)-nighttime |

The traditional temperature profiles shown are calculated using count profiles consisting of glued analog and digital measurements which are corrected for non-linearity and background before processing. The vertical resolution of the traditional temperature profiles is 30 m at the lowest heights, increasing to 400 m by the upper heights to decrease the magnitude of the statistical uncertainty. A calibration function linear in $1/T$ is used and the two coefficients have been determined with radiosonde

data using the altitude range from surface to 10 km. Hence, for this comparison the temperature profile is smoother than the



OEM retrieved temperature profile. The change in vertical resolution is based on the random uncertainty of the temperature profile at each height. The vertical resolution is decreased until the temperature uncertainty becomes less than a threshold value, set here as 1 K.

### 5.1 Case 1: Nighttime with clear conditions

5  Fig. 1 shows the RALMO 30 min coadded count measurements in the four PRR channels for case 1 given in Table 3. Analog signals suffer day and night from a high electrical offset which becomes dominant above 5 km. On the other hand, the analog signals stay linear over the entire encountered range. The digital signals on the other hand have a very low system background but become saturation above 2 MHz. Fig. 2 shows the residuals, which are defined as the difference between the forward model and the measurements. For a good retrieval with cost on the order of unity, the residuals (blue curve) should be on the order

10  of the standard deviation of the measurement noise (red curve), and indeed this is the case, hence the forward model is not over-fitting the measurements (e.g. cost much less than unity).

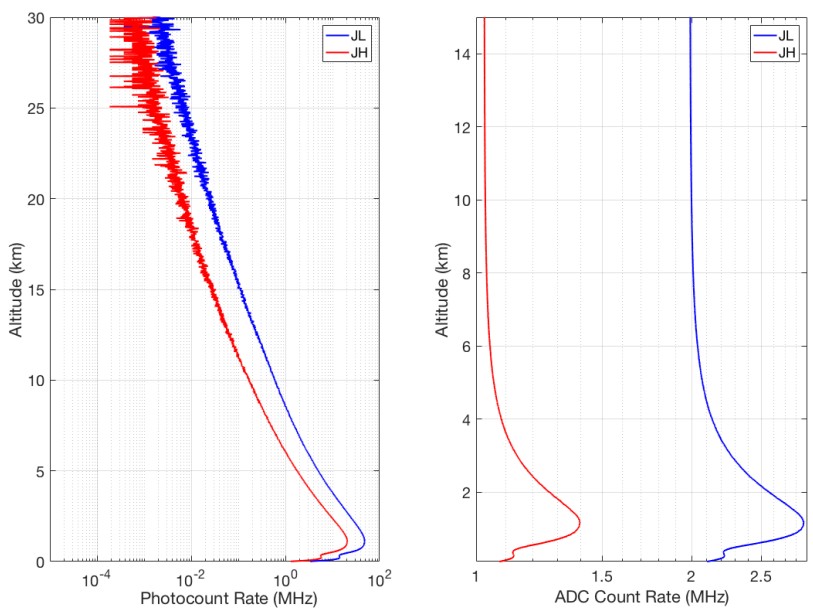

**Figure 1.** Count rate for 30 min of coadded RALMO measurements from 2300 UT on 09 September 2011, a clear night. Left panel: digital channels (blue curve, JL; red curve, JH). Right panel: analog channels.





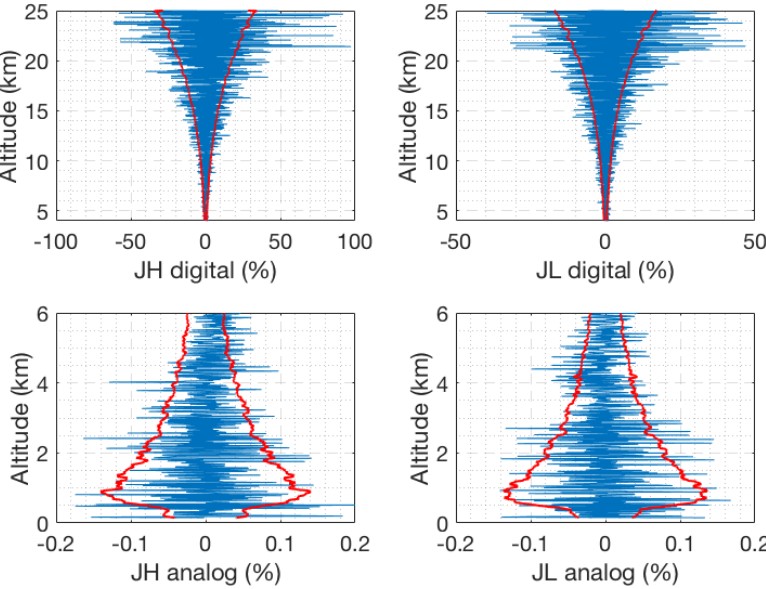

**Figure 2.** Difference between the forward model and clear nighttime RALMO measurements on 09 September 2011 for the four channels (blue). The red curves show the standard deviation of the measurements noise.

The averaging kernels of temperature and the vertical resolution of the retrievals are shown in Fig. 3. The area of the averaging kernels is defined as the response function of the retrievals and is close to 1.0 below 24 km, meaning that contribution of the *a priori* temperature profile to the retrieved temperature is negligible (Rodgers, 2000). With decreasing signal-to-noise ratio the measurement response quickly decreases above about 27 km (Fig. 3). The full-width at half-maximum of the averaging kernels is the retrieval's vertical resolution (Fig. 3, right panel). The vertical resolution of the retrieval is smaller than the resolution of the retrieval grid above about 10 km. We consider the height at which response function decreases to 0.9 as the cutoff height for the OEM retrievals as at this level the contribution of the *a priori* temperature profile is about 10%.



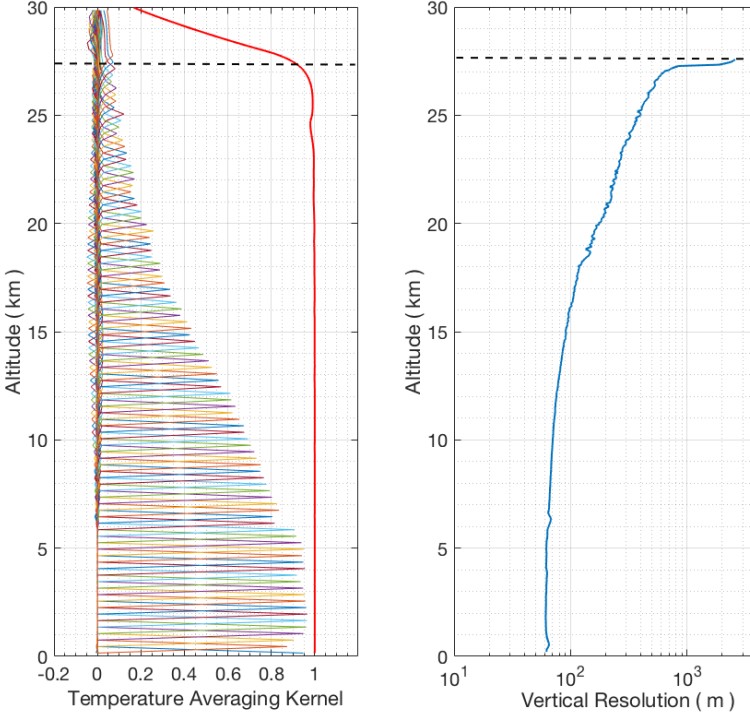

**Figure 3.** Averaging kernels (left) and vertical resolution (right) for temperature retrievals from the clear nighttime RALMO measurements on 09 September 2011. The area of the averaging kernels at each height, the response function, is the red solid line. The horizontal dashed line shows the height below which the retrieval is due primarily to the measurement and not the *a priori* temperature profile. For clarity averaging kernels for every fifth height bin of the retrieval grid are shown.



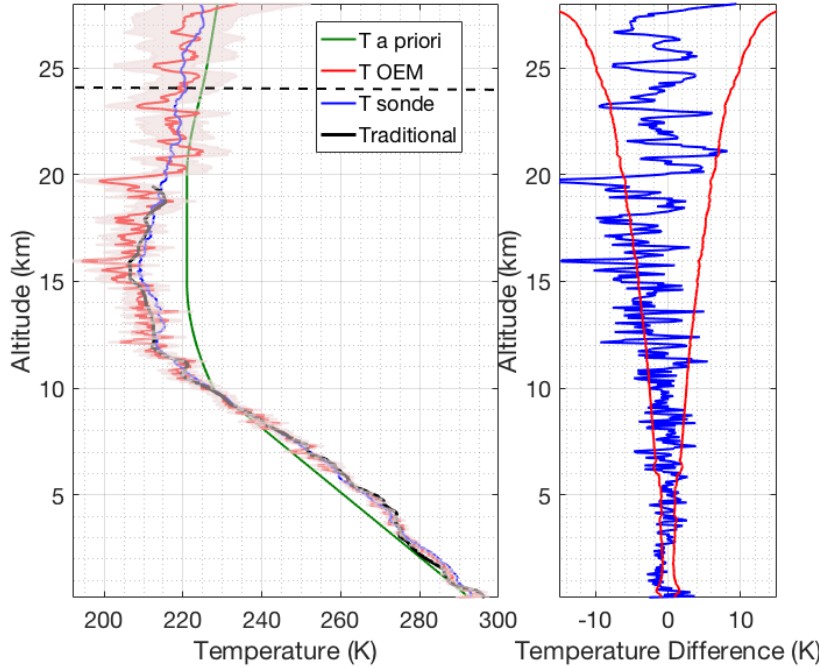

**Figure 4.** Left panel: Retrieved temperature profile and the statistical uncertainty (red curve and shaded area) using the OEM from the clear nighttime RALMO measurements on 09 September 2011. The blue curve is the radiosonde measurement. The sonde was launched at 2200 UT. The green curve is the *a priori* temperature profile used by the OEM. The black curve shows the temperature retrieved using the traditional analysis method from glued analog and digital signals. The horizontal dashed line shows the height below which the retrieval is due primarily to the measurement and not the *a priori* temperature profile. Right panel: The blue curve shows the temperature difference between OEM-retrieved temperature and the sonde temperature. The red curves in the figure show the statistical uncertainty of the OEM temperature.

Figure 4 shows a comparison of the OEM-retrieved temperature profile with coincident sonde temperature and temperature obtained using the traditional method. The traditional method profile from the RALMO glued (digital and analog) measurements provided by MeteoSwiss has a vertical resolution of 30 m, below 12.5 km and 400 m above this height, and is interpolated to the same retrieval grid as the OEM and shown in black. The change in vertical resolution and the cutoff height of the traditional temperature retrieval are based on temperature uncertainty thresholds. As shown in the right panel of Fig. 4, the temperature difference between OEM-retrieved and sonde temperature (blue curve) fits inside the statistical uncertainty of the OEM-retrieved temperature. Temperature from the traditional method follow the same trend as the sonde and the OEM-retrieved temperature.

The OEM provides a complete uncertainty budget of both random and systematic uncertainties (Figure 5). Uncertainties due to the Rayleigh cross section is in the order of 0.01 mK. Pressure accounts for up to 0.1 K below 10 km and up to 0.7 K up to





25 km and is a non-negligible source of uncertainty in the stratospheric part of the retrieval. Note that this error could be reduce by choosing a better pressure profile. The uncertainty due to the analog coupling constant $R_a$ is in the order of 0.07 K up to 4 km and the uncertainty due to digital coupling constant $R$ is 0.15 K in 4-7 km height range and less than 0.1 K everywhere else. The largest contribution to the temperature uncertainty is from measurement noise, which increases with height.

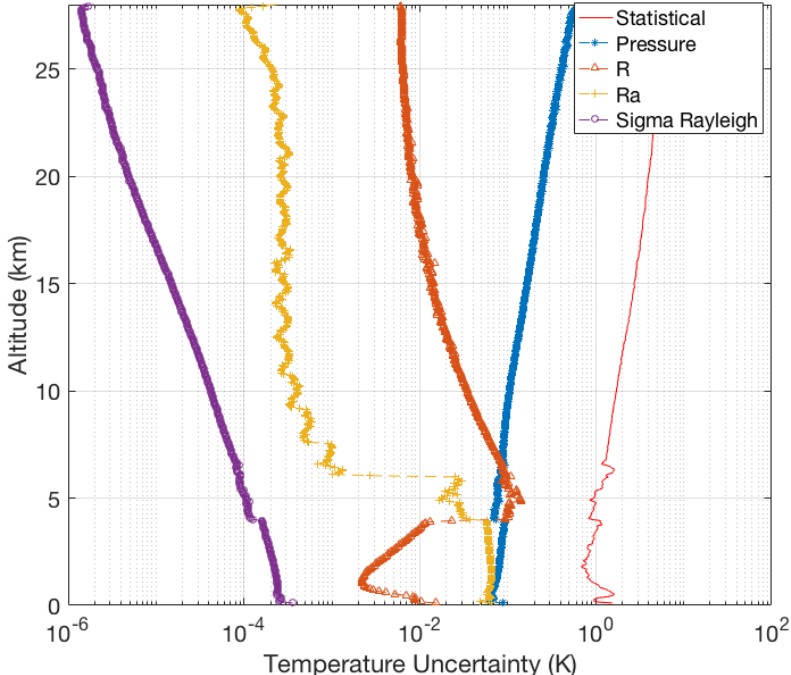

**Figure 5.** Random uncertainties (red curve) and systematic uncertainties due to the forward model parameters for the temperature retrievals from the clear nighttime RALMO measurements on 09 September 2011. The systematic uncertainties included are Rayleigh-scatter cross section (purple dot-dash curve), $R$ digital coupling constant (orange triangle-dash curve), $R_a$ analog coupling constant (yellow cross-dash curve), and pressure (blue dot-dash curve).



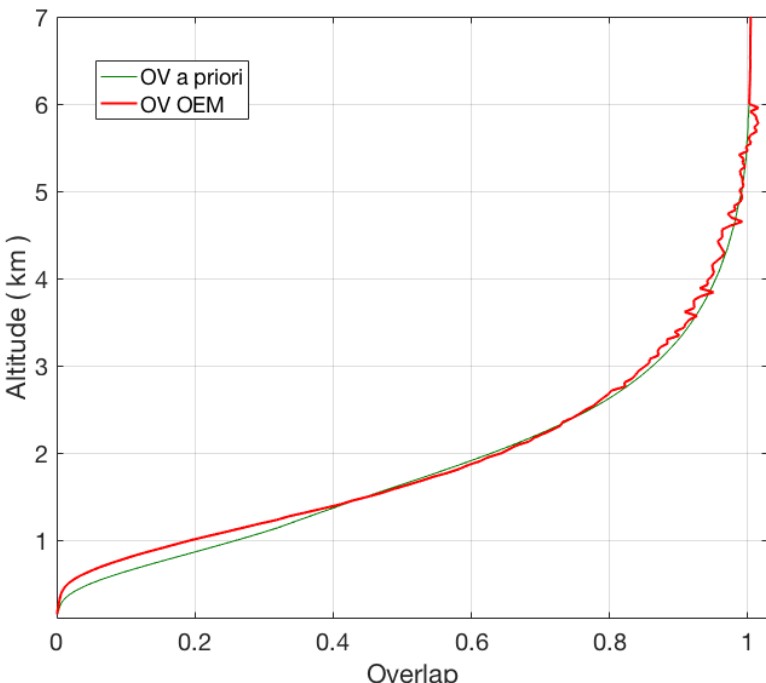

**Figure 6.** Retrieved geometrical overlap function (red curve) from the clear nighttime RALMO measurements on 09 September 2011 compared to the *a priori* geometrical overlap function (green curve).

Figure 6 shows the OEM-retrieved geometrical overlap function for the RALMO PRR channels. It illustrates that the overlap retrieval is constrained to be equal to 1 above the transition height (6 km), above which the extinction coefficient is retrieved (see Sec. 4.3).

## 5.2 Case 2: Daytime with clear conditions

5 The retrieval setup for second case study, which is a daytime measurement (Table 3), is identical to the one used for nighttime. The major difference between daytime and nighttime retrievals is the large solar background in the measurements, which is evident in the measurements of the digital PRR channels (Fig. 7).





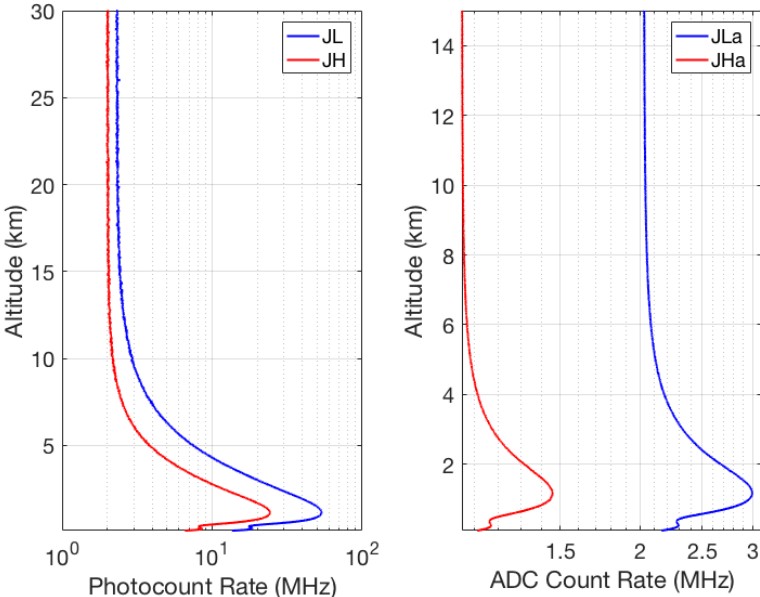

**Figure 7.** Count rate for 30 min of clear coadded RALMO measurements from 1100 UT on 10 September 2011. Left panel: digital channels (blue curve, JL; red curve, JH). Right panel: analog channels.

The residuals are unbiased and fall within the limits of the measurement standard deviation (Fig. 8). This result confirms the capability of our forward model in daytime conditions. As shown in Fig. 9, the vertical resolution (right panel) is the same as the retrieval grid below 13 km where response function (left panel) is equal to 0.9. The vertical resolution starts to deviate from the retrieval grid as the response function decreases and the background starts to dominate the digital channels.





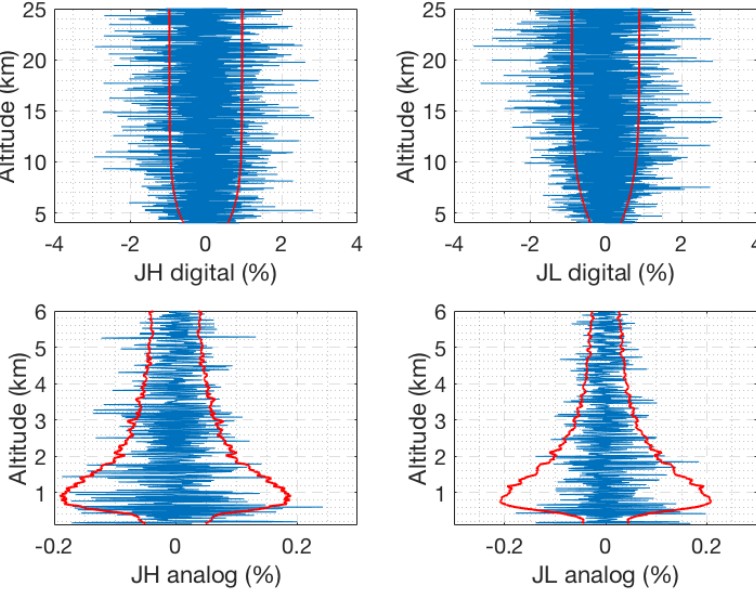

**Figure 8.** Difference between the forward model and the clear daytime RALMO measurements on 10 September 2011 for the four channels (blue). The red curves show the standard deviation of the measurements.

Similar to the clear nighttime case, the OEM-retrieved temperature agree with the sonde temperature within the statistical uncertainty (Fig. 10). It is also evident for this specific case study that the temperature from the traditional method deviate from the sonde temperature more than the OEM retrieved temperature. This difference compared to the traditional method is due to the fact that the OEM adapts the vertical resolution in an optimal way as a function of height while the traditional method is,
5  constrained by the filter employed to a specific signal-to-noise ratio of the measurements from both channels.





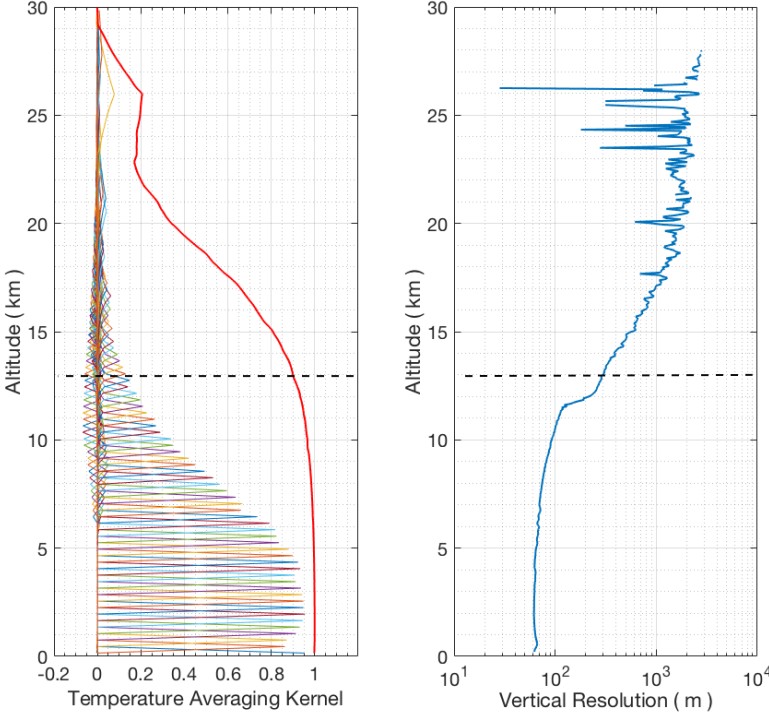

**Figure 9.** Same as Fig. 3 but for 10 September 2011.

The analog coupling constant $R_a$ uncertainty of the temperature from the daytime measurements are also in the order of 0.07 K and the uncertainty due to digital coupling constant $R$ is less than 0.2 K for all heights.

The retrieved geometrical overlap function for the clear daytime case (not shown) agrees with the geometrical overlap retrieved for the nighttime case within 10% statistical uncertainty.



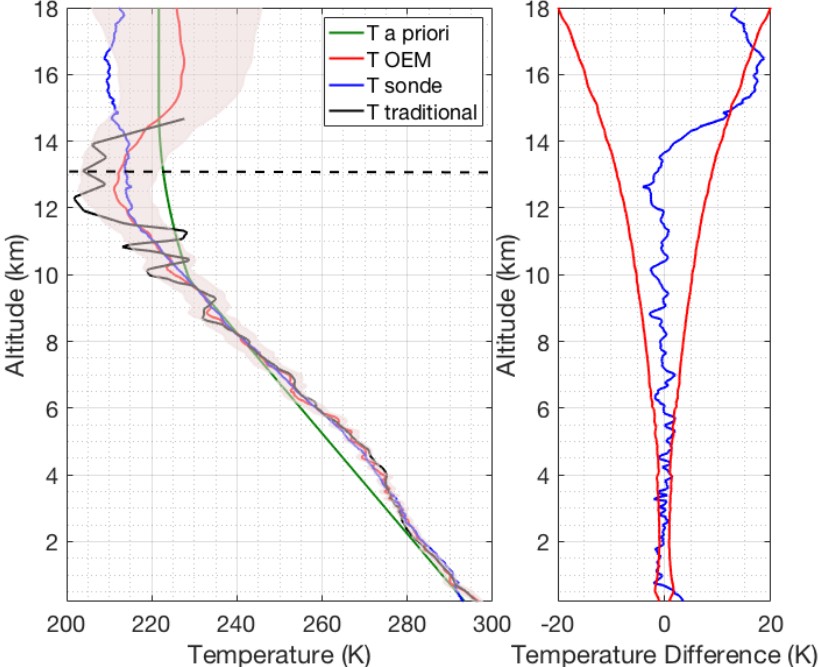

**Figure 10.** Same as Fig. 4 but for 10 September 2011.



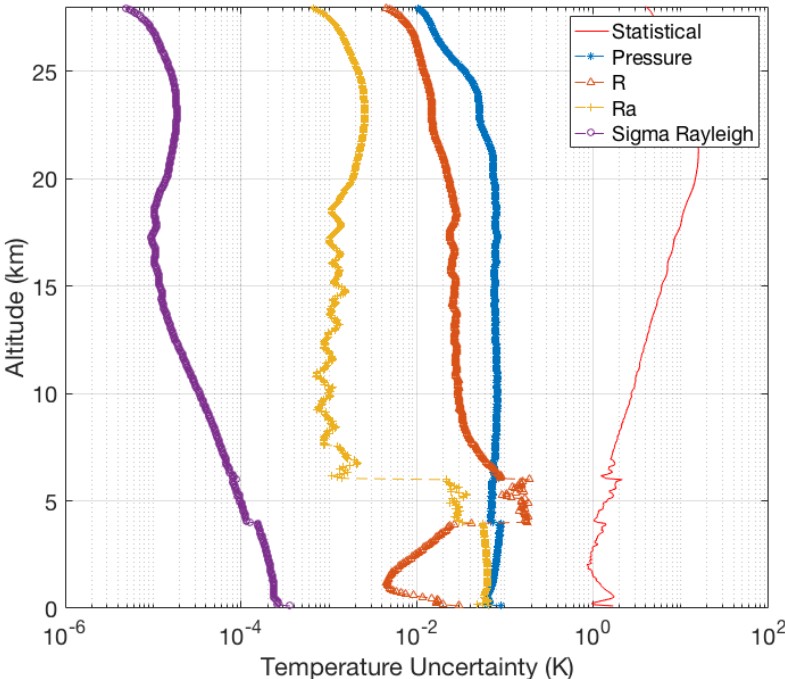

**Figure 11.** Same as Fig. 5 but for 10 September 2011.

## 5.3   Case 3: Nighttime with cirrus cloud

The third case (details are given in Table 3) features a cirrus cloud at 6 km height (Fig. 12). The retrieval setup is identical to the previous cases, as the cloud base is above the height of full geometrical overlap of the transmitter and receiver. The a priori profile of the particle extinction coefficient is derived from the backscatter ratio assuming a lidar ratio of 15 sr.





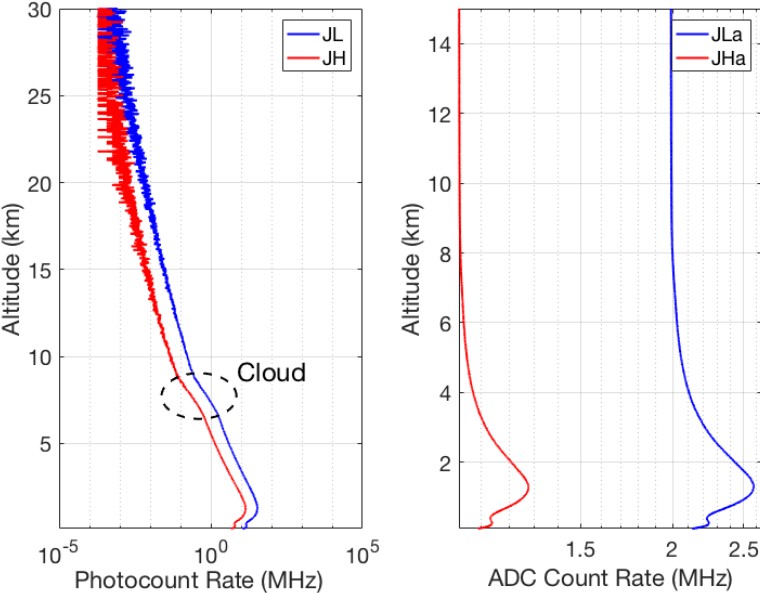

**Figure 12.** Count rate for 30 min of coadded RALMO measurements from 2300 UT on 05 July 2011. Left panel: digital channels (blue curve, JL; red curve, JH). Right panel: analog channels.

The residuals (Fig. 13) are unbiased and fall within the square root of the measurement variance. This is also true for the altitude range of the cirrus cloud demonstrating that the particle extinction coefficient was determined correctly. The response function (left panel, Fig. 14) decreases to the 0.9 cutoff value at about 23.5 km, clearly lower than the clear-sky nighttime case because of the attenuation of the cirrus cloud. Similar to the two previous cases, the OEM-retrieved temperature agree with the

5    sonde temperature within the statistical uncertainty of the OEM retrieved temperature (Fig. 15).



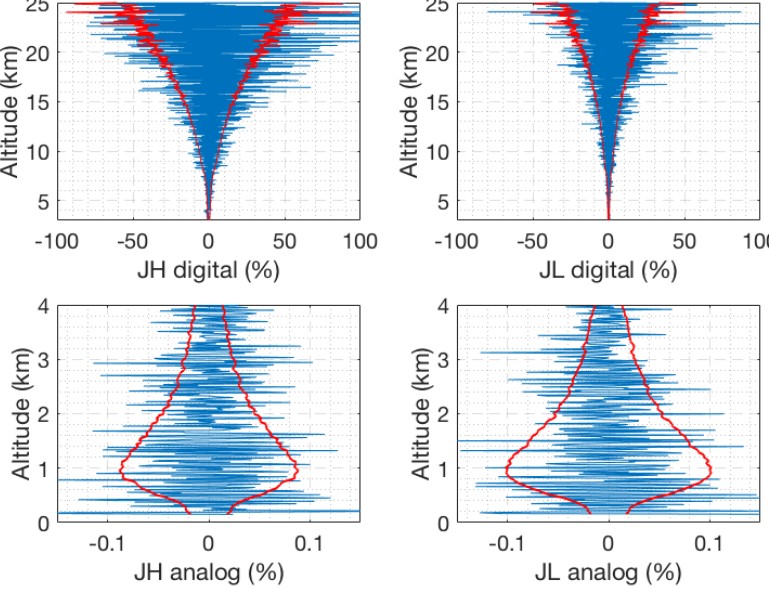

**Figure 13.** Difference between the forward model and the nighttime RALMO measurements on 05 July 2011 with the presence of a cirrus cloud for the four channels (blue). The red curves show the standard deviation of the measurements.



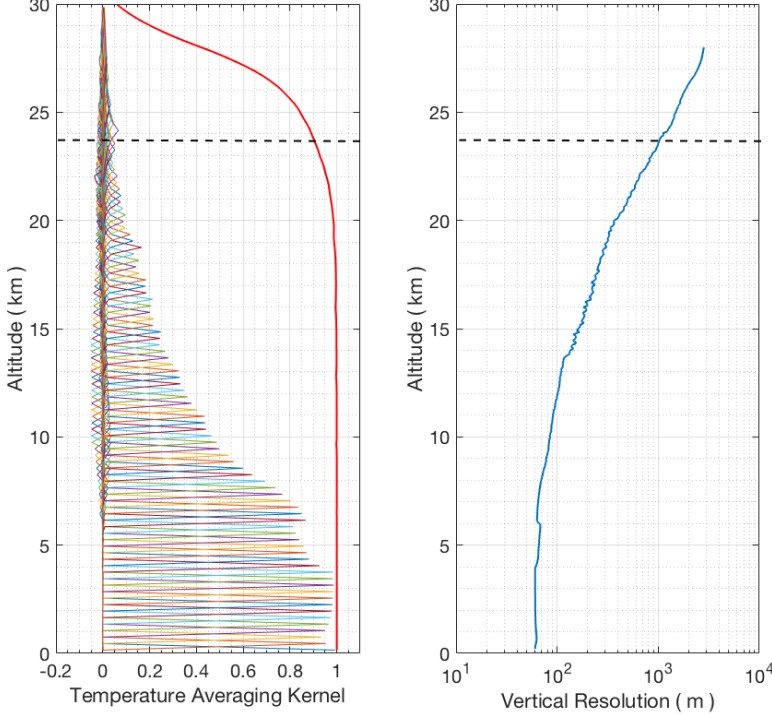

**Figure 14.** Same as Fig. 3 but for 05 July 2011 with a cirrus cloud at 6 km height.



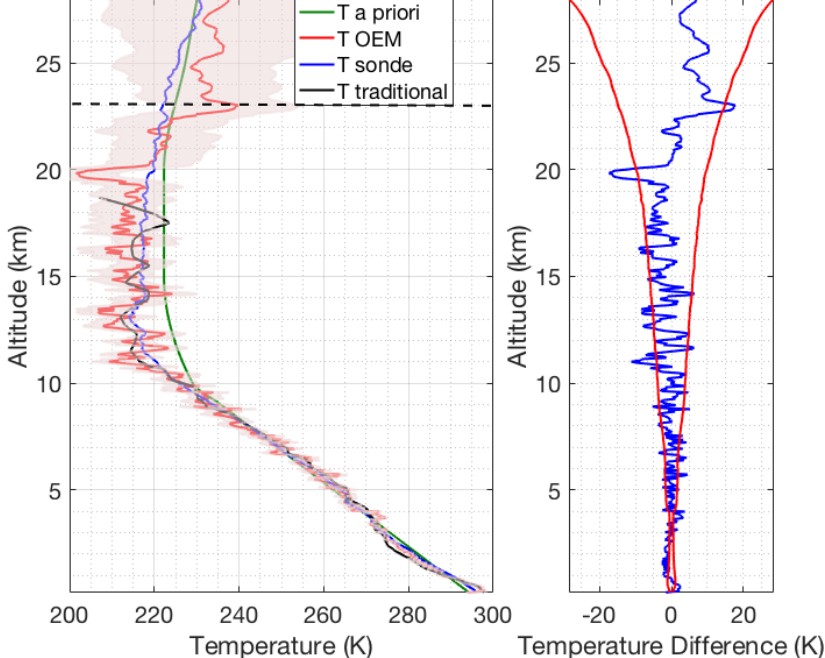

**Figure 15.** Same as Fig. 4 but for 05 July 2011 with a cirrus cloud at 6 km height, using the OEM.



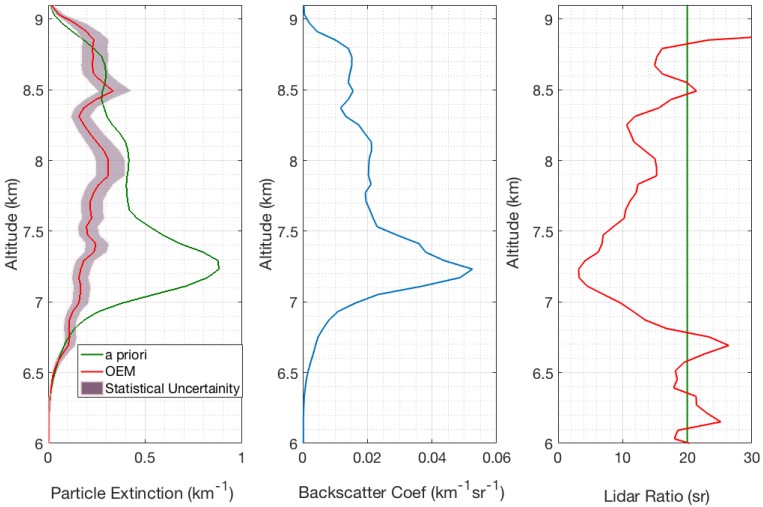

**Figure 16.** Left: Retrieved particle extinction (red) and the *a priori* particle extinction used in the OEM (green). Center: Backscatter coefficient calculated from the nighttime RALMO measurements on 05 July 2011 with of a cirrus cloud present at 6 km height. Right: Lidar ratio used to determine *a priori* particle extinction (green) and the estimated lidar ratio using the OEM-retrieved particle extinction (red).

The retrieved geometrical overlap function from the measurement with the cirrus cloud (not shown) agrees within 10% uncertainty with the geometrical overlap functions retrieved from the measurement with clear sky conditions, as the cloud is above the region of complete geometrical overlap. The red curve in the first plot in Fig. 16 shows the OEM-retrieved particle extinction and the green curve is the *a priori* particle extinction estimated using the RALMO PRR and elastic measurements,

assuming a lidar ratio for cirrus clouds in order to estimate an *a priori* extinction (Section 4.4.3). Above 6.75 km, the OEM-retrieved particle extinction is around $0.25 \text{km}^{-1}$ and approximately two times smaller than the *a priori* yielding a lidar ratio of 5-15 sr while the initial guess was 20 sr (Fig. 16. Ansmann et al. (1992b), using independent measurements of particle extinction and backscatter profiles in cirrus clouds, show similar extinction values ($0 - 0.5 \text{km}^{-1}$) and also similar values for the lidar ratio inside the cloud 0 - 10 sr. Thus, the OEM-retrieved extinctions for this cirrus cloud appears to be reasonable.Below

6.75 km, the lidar ratio is around 20 sr which could be and indication that this part of the cloud is super-cooled liquid.

The uncertainty budget for this case (not shown) is similar to the previous 2 cases shown; the cloud has little effect on the uncertainty values. As before, the statistical uncertainty makes the largest contribution to the full uncertainty.

## 5.4 Case 4: Nighttime with lower level cloud

A cloud at about 4 km is present in measurements used for the last case study (Table 3). In this situation we use our OEM-

15 retrieved geometrical overlap during clear conditions as our *a priori* geometrical overlap profile. We then retrieve geometrical





overlap to the cloud base (4 km height) and particle extinction above 4 km. In this case the retrieved geometrical overlap up to
km agrees within 10% uncertainty with the OEM-retrieved geometrical overlap for clear conditions.

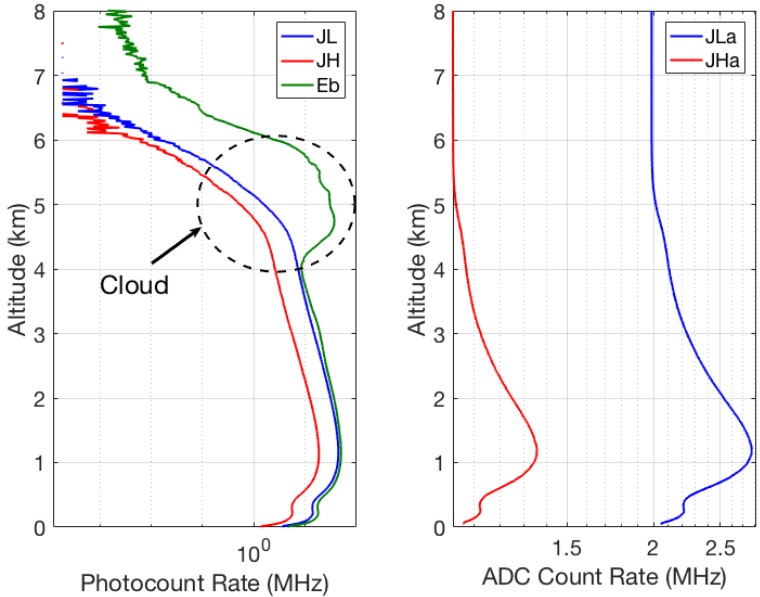

**Figure 17.** Count rate for 30 min of coadded RALMO measurements from 2300 UT on 21 June 2011, which has a cloud base at an height
about 4 km. Left panel: digital channels (blue curve, JL; red curve, JH; green, Elastic). Right panel: analog channels.





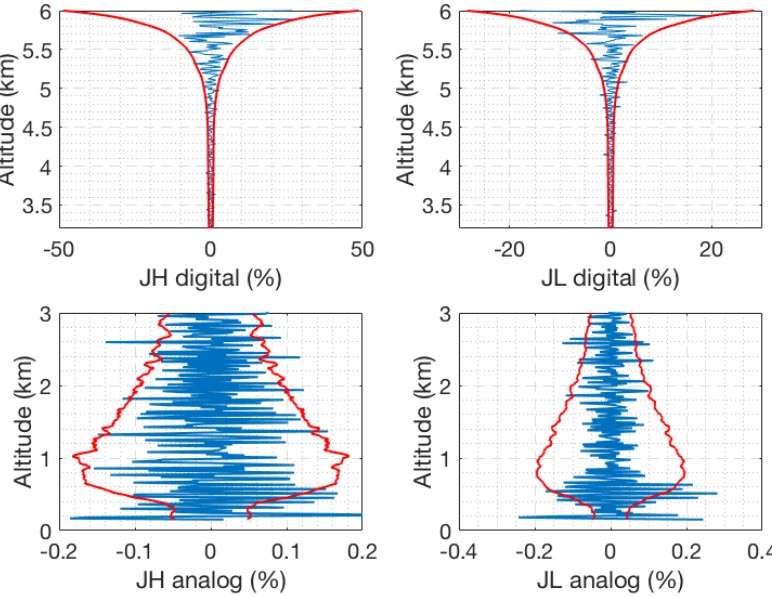

**Figure 18.** Difference between the forward model and the nighttime RALMO measurements on 21 June 2011 with the presence of a lower level cloud for the four channels (blue). The red curves show the standard deviation of the measurements.

Figure 17 shows the measurements in the four PRR channels and the elastic channel measurement (left panel, green curve). It can be seen in the elastic signal that a cloud base is at around 4 km height. The Raman measurements drop above 4 km and are fully attenuated at 7 km. We use these measurements obtained at a cloudy condition as input to our OEM, and obtain unbiased residuals which fall within the standard deviation of the measurements, meaning the forward model accurately retrieve temperature in the presence of the cloud.

The response function (left panel, Fig. 19) is 0.9 at 6 km, which is considered the maximum height where the OEM-retrieved temperature are valid. At this height the vertical resolution (right panel, Fig. 19) rapidly increases as the cloud thickens. As shown in Fig. 20 (right panel), up to 6 km, the temperature from the sonde launched at 2300 UT from Payerne and OEM temperature agree with each other within the statistical uncertainty of the OEM temperature. Temperature retrieved using the traditional method are similar to the OEM and sonde measurements up to 3.5 km, while inside the cloud the traditional temperature starts to deviate.





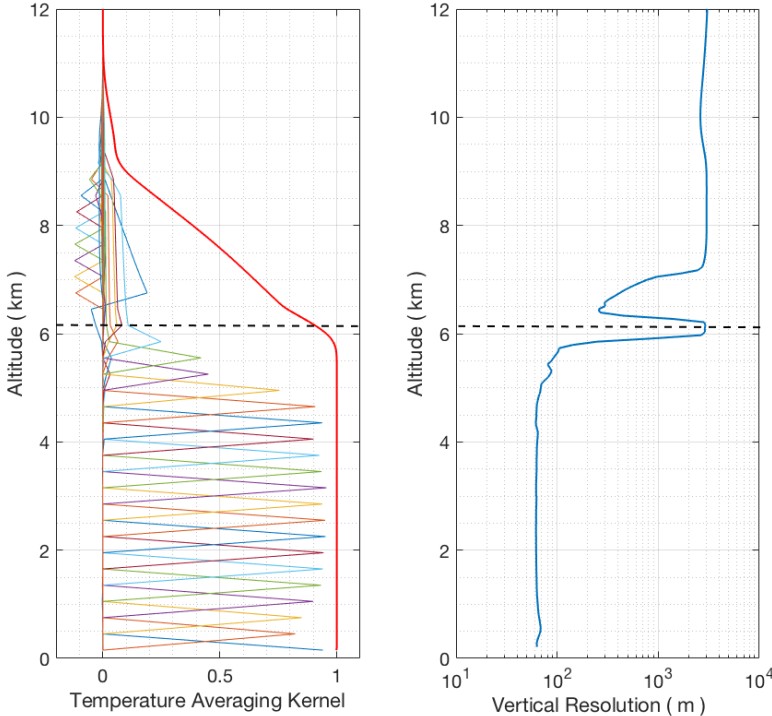

**Figure 19.** Same as Fig. 3 but for 21 June 2011 with the presence of lower level cloud.



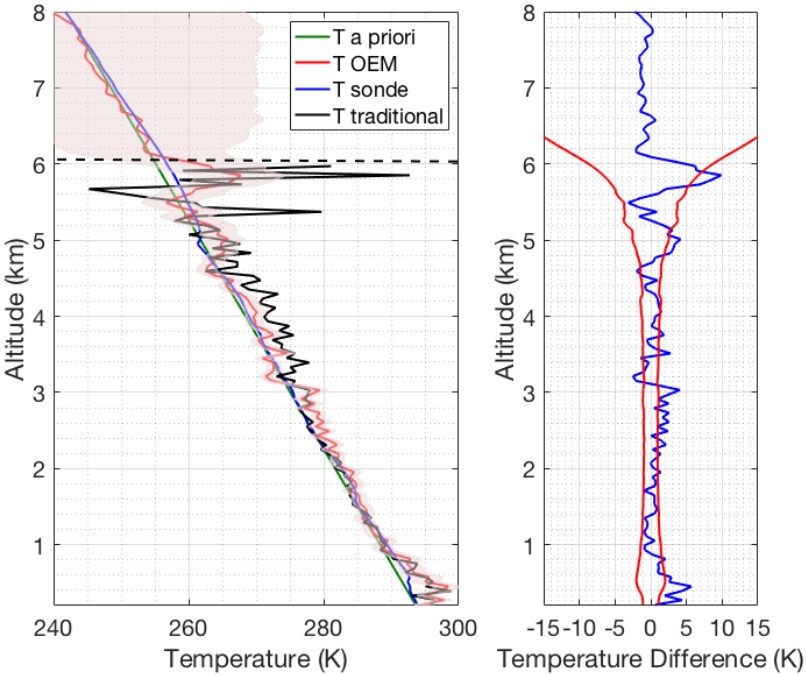

**Figure 20.** Same as Fig. 3 but for 21 June 2011 with the presence of a lower level cloud, using the OEM.

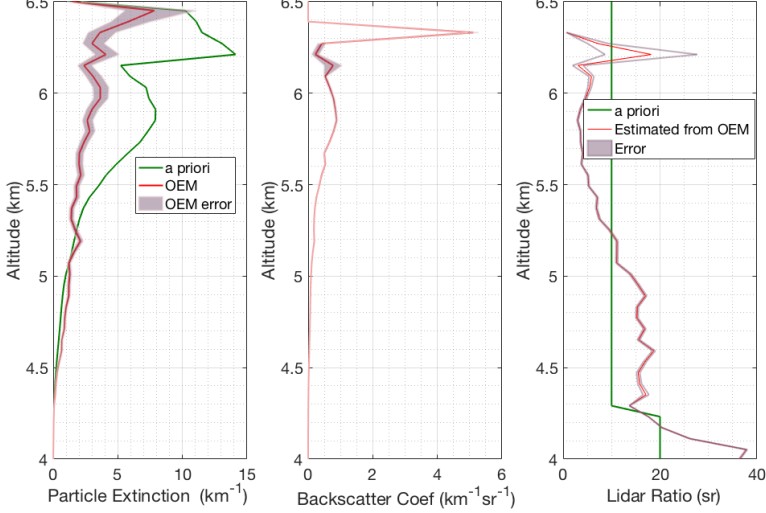

**Figure 21.** Same as Fig. 16 but for 21 June 2011 with the presence of a lower level cloud.





Below 5.25 km the retrieved particle extinction coefficient agrees well with the a priori values and the corresponding lidar ratio is between 15 and 20 sr indicating a liquid cloud. Above 5.25 km the retrieved particle extinction is smaller than the first guess yielding again a lidar ratio around 5 sr (left panel in Fig. 21). This could be an indication that the cloud became an ice cloud above 5.25 km.

## 6   Discussion

The four retrievals discussed in the previous section demonstrate that the OEM provides robust and accurate retrievals of temperature, geometrical overlap, and particle extinction coefficients, during both clear and cloudy day and night conditions. Unlike the traditional Raman lidar temperature analysis method (Cooney, 1972; Arshinov et al., 1983; Di Girolamo et al., 2004; Behrendt, 2005; Zuev et al., 2017), the OEM does not require an analytic form of a calibration function; rather a single
calibration coefficient has to be estimated using a reference temperature profile and were shown to have a small effect on the retrieved temperature.

The calibration function plays a key role in the traditional temperature retrieval algorithm from the PRR backscattered signals, in particular if calibration is not done over a the entire observed temperature range. Typically, a calibration function linear in $1/T$ is used for systems that detect only one or multiple RR lines (Behrendt, 2005), although other forms of calibration
functions have been employed. Recently Zuev et al. (2017) showed closer agreement at times with temperature calculation used a higher order polynomial for the calibration function. All calibration coefficient estimation methods require multiple reference data points which span ideally the entire range of temperatures to avoid extrapolation errors.

The only calibration required in our OEM scheme is the determination of the two coupling constants, $R$ and $R_a$. The coupling constants can be estimated at a specific height (that is over a narrow range of temperature) without introducing extrapolation
errors. Using the OEM we can show that the contribution of the coupling constant to the temperature uncertainty is in the order of 0.07 K in the height below 4 km and about 0.2 K or less above 4 km for a wide variety of sky conditions.

The OEM temperature retrievals of 4 very different sky conditions have been compared against coincident radiosonde temperature measurements. cases presented is the US Standard model normalized to the surface temperature from the coincident sonde temperature. We successfully used other *a priori* temperature profiles, such as the smoothed sonde temperature measure-
ments and temperature from the Mass Spectrometer and Incoherent Scatter radar (MSIS) model to retrieve temperature using our OEM algorithm. All the retrieved temperature profiles using each *a priori* profile for heights where the response function is 0.9 or greater are identical within the statistical uncertainty.

In our study, we have successfully retrieved a geometrical overlap function for the RALMO system using the PRR measurements simultaneously with the temperature retrieval. Ray-tracking studies have concluded that the RALMO system reaches
its full geometrical overlap by 5.0-5.5 km in height. These calculations are consistent with our geometrical overlap retrievals in both clear daytime and nighttime conditions. Measuring the geometrical overlap function and its uncertainty allows a more accurate estimation of the particle extinction coefficient when clouds or aerosols are present. The particle extinction profiles



we retrieved in the two cloudy condition cases are consistent with measurements collected by Ansmann and Müller (2005) for cirrus clouds and with O'Connor et al. (2004) for liquid clouds.

The particle extinction coefficient is retrieved in the full geometrical overlap region, i.e. above 6 km or above the cloud base. The extinction values and lidar ratios we obtained for high and mid-level clouds are in agreement with other publications. The

two case studies featuring clouds suggest, that both clouds consisted of a liquid and a ice part with lidar ratios at 18 and 5 sr.

For all the case studies we presented, the lidar constants for the lower quantum channels (analog and digital), the dead times for each digital channel (JL and JH), and background for all four channels are also retrieved. The retrieved lidar constants for each channel agreed within 20% uncertainty for all four cases. The retrieved dead times are about 3.8 ns and consistent with the dead times specified by the manufacturer and other independent estimates.

The uncertainty budget provided by the OEM contains both random and systematic uncertainties. Estimation of the uncertainty budget requires assignment of appropriate covariances to the model parameters. Using the standard deviations given in Table 2, the uncertainty budgets for all the case studies are estimated. The largest contribution towards the temperature uncertainty originates from the statistical uncertainty due to the measurement noise. Overall contribution from the coupling constants to the temperature uncertainty are less than 0.2 K for all heights. Given the fact that the measurement noise can be reduced with

longer integration times this result suggests that the OEM method very precise temperature measurements are possible even if calibration is only possible over a small temperature range. The systematic uncertainties of pressure and air density are on the order of 0.1 K to 0.1 mK respectively. Understanding the full uncertainty budget of temperature is of particular importance for trend analysis and process studies. The observational basis for super-saturation studies in the upper troposphere is still unsatisfactory and the OEM framework allows to combine different data sources to provide a high quality data set including

profile-by-profile uncertainty budgets. .

# 7  Conclusions

We have demonstrated the ability of the OEM to retrieve multiple geophysical and instrumental parameters from PRR lidar measurements. The first-principle forward model adequately represents the raw PRR measurement and allows us to retrieve temperature, geometrical overlap, particle extinction, lidar constants, background counts, and dead time using multiple analog

and digital channels. The retrievals discussed for four different cases that represent different (and typical) sky conditions. We found the following results from our OEM temperature retrievals from PRR measurements:

– The forward model presented, based on the lidar equation, contains the essential physics to reproduce the analog and digital measurements, leading to unbiased residuals and robust estimates of temperature.

– Our OEM retrieval does not require a calibration function as used in the traditional temperature retrieval method. It

only requires determination of the two coupling constants, $R$ and $R_a$, using a reference temperature profile that can be estimated at a specific height bin (or over a range). Retrieved temperature profiles from both day and night uncorrected PRR measurements in clear and cloudy conditions agree well with coincident radiosonde measurements.





- The OEM provides a cutoff height for the temperature retrievals that specify up to which height the retrieved profile is primarily due to the measurements and not the *a priori* temperature profile.

- Vertical resolution is determined at each height, and is automatically adapted in the retrieval in response to increasing measurement noise with height.

- The OEM provides a complete uncertainty budget, including random and systematic uncertainties due to model parameters, including the assumed pressure, air density and the coupling constants.

- Simultaneous analog, which are linear, and digital measurements allow the dead time to be retrieved.

- The OEM-retrieved geometrical overlap function for the RALMO using the measurements in clear conditions is determined and shown to be consistent with, but not the same, as that calculated by Dinoev et al. (2013). Hence, retrievals
of the particle extinction coefficient are possible using the OEM from the measurements in cloudy conditions or when aerosol layers are present.

- The OEM is a computationally fast and practical for routine temperature retrievals from lidar measurements as required for operational lidar systems.

We have demonstrated that the OEM allows retrieval of temperature from Pure Rotational Raman lidar measurements that
are consistent with the coincident sonde temperature. We discussed the advantages of the OEM over the traditional temperature retrieval algorithm. We can use the OEM-retrieved temperature to study temperature trends with the benefit of a full uncertainty budget provided by our OEM. Our OEM temperature retrieval can also be used for routine measurements in a wide variety of observing conditions, and is applicable to any similar PRR lidar system.

We are in the process of implementing the OEM for routine temperature measurements from the RALMO system. We
are also combining the OEM PRR temperature retrieval with the OEM water vapor mixing ratio retrieval of Sica and Haefele (2016) to directly retrieve relative humidity from the RALMO measurements, both for its importance to operational forecasting and to allow the study of ice super-saturation events. We are also assimilating ERA5 hourly reanalysis data into OEM relative humidity and temperature algorithm to improve the accuracy of the OEM relative humidity retrievals.

*Acknowledgements.* We thank Prof. V. Simeonov (École Polytechnique Fédérale de Lausanne) for assisting us with the interpretation of the
RALMO measurements and details of the instrumentation. We thank Ghazal Farhani for her helpful comments and suggestions that were extremely useful to us. We also thank the Western writing support center and Patricia Sica for their assistance in editing and proofreading this paper. This project has been funded in part by the National Science and Engineering Research Council of Canada and by the Canadian Space Agency under the Arctic Validation and Training for Atmospheric Research in Science (AVATARS) program.





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
