# Peer review of "Retrieval of Temperature From a Multiple Channel Pure Rotational Raman-Scatter Lidar Using an Optimal Estimation Method"

_Atmospheric Measurement Techniques, 2019_

## Referee Comment (RC1) · Christoph Ritter (Referee) · 3 Jun 2019

Review of „Retrieval of Temperature from … by optimal estimation method"  by Mahagammulla Gamage, et al.

The paper describes the temperature retrieval by lidar using the optimal estimation method. Four different cases (each day / night and clear / cloudy) from Payerne station are used to demonstrate the feasibility of the method. While the OEM method for temperature retrieval is not entirely new and the first and third bullet point of the conclusions may be found elsewhere, the paper as a whole contents enough information for publishing. It is generally well written and well structured. The 4 selected cases are new and demonstrate the usage of the still not too far spread OEM technique.

I have a short list of questions and remarks to the author which should be clarified prior to publishing:

You show a complete error analysis for the OEM technique, which I found convincing. However, to better judge these results it would be very interesting to have a similar error estimation for the traditional Raman technique. Especially in your case 1 fig. 4 it seems as if the Raman result is only plotted up to 12 km although in the manuscript a change of vertical resolution in this altitude is mentioned (page 11). Can you comment on this?

Do you assume the same overlap for both channels? Why do you need the overlap? Do you really think to be able to retrieve the overlap with the required precision to obtain aerosol extinction information? In this case an error analysis would be required, otherwise revise your wording. I think for this paper such an effort is not necessary – as the particle extinction is the same for the high and low channel, hence the temperature from the OEM should not depend on the extinction?

Page 11 and Fig. 5: your error analysis is nice and one of the strong selling points of this paper. However, in the current form I cannot reproduce the values. A bit more information is required, how the values were obtained.

Minor points:

Page 1 quote Mahagammulla Gamage: I think in the introduction it is not necessary to quote a paper which is still under preparation. You may choose another quote here

Page 3: (and elsewhere) is d sigma / d Omega really the ATTENUATED cross section? I am not sure, as you have Gamma^2 as extinction term in your eq. 1.

Page 4: explain x_a in eq. 7.

Page 5: minus sign in eq. 8 is missing

Page 9: 2 times "from" in line 14

Page 10, fig. 1: is the units of your analog signal MHz, how did you convert it?

Page 11: line 13: reduced

Page 13: line 4: agrees, line 4: deviates

Page 19: a lidar ratio of 5sr is already quite small, can you estimate an error for the lidar ratio?

Page 20: line 8: "by" missing … that by the OEM method; line 9: I don't understand the "temperature range" – I thought the OEM only depends on K and K_a?

Page 21: line 11: what do you mean be "uncorrected" PRR measurements?

---

## Referee Comment (RC2) · Christoph Ritter (Referee) · 8 Jul 2019

The authors answered all my questions in a satisfactory way. Hence, I have no objections to publish the article in AMT.

---

## Referee Comment (RC3) · Anonymous Referee #1 · 23 Jul 2019

The authors present the application of the OEM to pure rotational Raman (PRR) temperature lidar. The OEM has been applied before to other lidar techniques but not to this one. Compared to how PRR temperature lidar data were analysed previously, the OEM shows several advantages, the largest of which is that no calibration with other sensors is needed. Furthermore, the systematic and statistical uncertainties as well as the effect of partial overlap in the near range are also determined.

The authors could highlight these advantages even more clearly. There is one early study on calibrating RRL temperature lidar with the instrumental parameters (Vaughan

et al.: Atmospheric Temperature Profiles Made by Rotational Raman Scattering. Applied Optics, 1993) but this study concluded that the uncertainties were too high at that time.

The paper is mostly well written (see detailed below). My recommendation is to accept the paper after minor revision.

Specific comments:

Since the proposed technique seems to be universally applicable to temperature rotational Raman lidar systems, more references to the state-of-the-art of this technique and other existing systems would certainly be interesting for the readers and should be included.

You mention two assumptions on page 8, line 25/26. What do you mean with "well known"? How critical are these assumptions? When highlighting which assumptions are NOT needed (see abstract, introduction, conclusions), you should be fair and also mention which are needed. Maybe it would be interesting to explain also in the abstract and conclusions that (how much) the results are independent from the selected a priori temperature profile.

Page 9, line 4: There are always aerosols in the troposphere. How few are acceptable? I think the term "digital measurements" for photon counting signals is odd. Also the analog signals are digitized. Thus, I suggest that you write "photon counting" throughout the text.

Minor comments:

"Lidar constants" and "coupling constants" are terms which are not commonly used. Please define/explain or avoid. E.g., page 1, line 5: "ratio of efficiencies" is better I think since the laser power, transmitter efficiency, telescope area are the same for both channels and thus cancel and become irrelevant (if I understand correctly).

Page 1, line 21: "...2 flights per day". One could add: "only at selected sites world-
wide".

Page 1, line 22: Please add references to other combined Raman lidar systems. There are several which measure water vapor and temperature.

Page 3, Eq. 1: You assume that all PRR lines of one channel are collected with the same efficiency. This is generally not the case but may be true for RALMO. Please add a comment on this.

Page 3, Eq. 2: This equation is not found in Penney et al. 1974.

Page 5, line 11: Why is the background noise B\_RR a function of height? I think the "real background" should be height independent. If the baseline is height dependent, detector non-linearity or electronic cross-talk is present, I guess.

Page 7, table 2: What do you with "transition height"?

Page 8, line 6: beta\_par \*is\* related

Page 9, line 20: I would prefer "model parameters" or "model b parameters"

Page 9, line 21: Please refer to the figure.

Page 10: line 6: What do you mean with "dominant"?

Page 10, line 7: How do you know that the analog signals are linear?

Page 10, line 8: "become saturated". What do you mean with "saturated"?

Page 11, Fig 2: I suggest that you write "four signals". It is \*two\* channels. How did you determine "measurement noise"? With OEM? Please clarify.

Page 14, line 2: What is coupling constant R\_a? How is it defined?

Page 15, Fig 6: How did you obtain the a priori overlap? Why is this important since the ratio of the overlap functions cancels?

Figs 1, 7, 12, 17: I would prefer the same scales. JLa -> JL, JHa -> JH? Should be
consistent. I think it would be interesting to show the elastic signals for all cases. Why "Eb" for "elastic"?

The language still needs polishing/corrections. Here two examples:

Page 1, line 3ff: "assumption for the form of ...." However, I think the form of the calibration function is not really the point here. Calibration with external sensors (with all the uncertainties related to the accuracy of the reference sensor and to the sampling differences) is usually needed but overcome with OEM.

Page 1, line 13 & 15: "under different sky conditions", "under clear and cloudy conditions"

---

## Author Response (AR1)

The authors present the application of the OEM to pure rotational Raman (PRR) temperature lidar. The OEM has been applied before to other lidar techniques but not to this one. Compared to how PRR temperature lidar data were analysed previously, the OEM shows several advantages, the largest of which is that no calibration with other sensors is needed. Furthermore, the systematic and statistical uncertainties as well as the effect of partial overlap in the near range are also determined.

The authors could highlight these advantages even more clearly. There is one early study on calibrating RRL temperature lidar with the instrumental parameters (Vaughan

et al.: Atmospheric Temperature Profiles Made by Rotational Raman Scattering. Applied Optics, 1993) but this study concluded that the uncertainties were too high at that time.

The paper is mostly well written (see detailed below). My recommendation is to accept the paper after minor revision.

Specific comments:

Since the proposed technique seems to be universally applicable to temperature rotational Raman lidar systems, more references to the state-of-the-art of this technique and other existing systems would certainly be interesting for the readers and should be included.

You mention two assumptions on page 8, line 25/26. What do you mean with "well known"? How critical are these assumptions? When highlighting which assumptions are NOT needed (see abstract, introduction, conclusions), you should be fair and also mention which are needed. Maybe it would be interesting to explain also in the abstract and conclusions that (how much) the results are independent from the selected a priori temperature profile.

Page 9, line 4: There are always aerosols in the troposphere. How few are acceptable? I think the term "digital measurements" for photon counting signals is odd. Also the analog signals are digitized. Thus, I suggest that you write "photon counting" throughout the text.

Minor comments:

"Lidar constants" and "coupling constants" are terms which are not commonly used. Please define/explain or avoid. E.g., page 1, line 5: "ratio of efficiencies" is better I think since the laser power, transmitter efficiency, telescope area are the same for both channels and thus cancel and become irrelevant (if I understand correctly).

Page 1, line 21: "...2 flights per day". One could add: "only at selected sites world-

wide".

Page 1, line 22: Please add references to other combined Raman lidar systems. There are several which measure water vapor and temperature.

Page 3, Eq. 1: You assume that all PRR lines of one channel are collected with the same efficiency. This is generally not the case but may be true for RALMO. Please add a comment on this.

Page 3, Eq. 2: This equation is not found in Penney et al. 1974.

Page 5, line 11: Why is the background noise $B_{RR}$ a function of height? I think the "real background" should be height independent. If the baseline is height dependent, detector non-linearity or electronic cross-talk is present, I guess.

Page 7, table 2: What do you with "transition height"?

Page 8, line 6: beta_par *is* related

Page 9, line 20: I would prefer "model parameters" or "model b parameters"

Page 9, line 21: Please refer to the figure.

Page 10: line 6: What do you mean with "dominant"?

Page 10, line 7: How do you know that the analog signals are linear?

Page 10, line 8: "become saturated". What do you mean with "saturated"?

Page 11, Fig 2: I suggest that you write "four signals". It is *two* channels. How did you determine "measurement noise"? With OEM? Please clarify.

Page 14, line 2: What is coupling constant $R_a$? How is it defined?

Page 15, Fig 6: How did you obtain the a priori overlap? Why is this important since the ratio of the overlap functions cancels?

Figs 1, 7, 12, 17: I would prefer the same scales. JLa -> JL, JHa -> JH? Should be

[Figure]

consistent. I think it would be interesting to show the elastic signals for all cases. Why "Eb" for "elastic"?

The language still needs polishing/corrections. Here two examples:

Page 1, line 3ff: "assumption for the form of . . ." However, I think the form of the calibration function is not really the point here. Calibration with external sensors (with all the uncertainties related to the accuracy of the reference sensor and to the sampling differences) is usually needed but overcome with OEM.

Page 1, line 13 & 15: "under different sky conditions", "under clear and cloudy conditions"
* * *
[Figure]

AMT-2019-107:  Retrieval of Temperature From a Multiple Channel Pure Rotational
Raman-Scatter Lidar Using an Optimal Estimation Method (Gamage et al.)
Response to Referee 2 (**Anonymous Referee #1**)
26 September 2019

The Referee's comments are in blue, are responses are in black. Red sections have been added to the manuscript.

Since the proposed technique seems to be universally applicable to temperature rotational Raman lidar systems, more references to the state-of-the-art of this technique and other existing systems would certainly be interesting for the readers and should be included.

We will add the following paragraphs to the introduction of the paper:

Behrendt (2005) provides a comprehensive overview of the traditional rotational Raman lidar temperature calculation method. Over the years the traditional temperature method has been improved by advancing the instrumentation capabilities and by improving the estimation and calibration techniques. Here are some examples of innovations in this area since the Behrendt (2005) review.

Studies by Radlach et al. (2008) and Weng et al. (2018) introduce changes to the individual Raman lidar systems to improve the temperature measurements. Radlach et al (2008)  introduced a new high-resolution rotational Raman lidar system with a receiving system that uses multicavity interference filters in a sequential setup to improve the efficiency of the elastic and rotational Raman signal separation. Together with the filter adjustments they have made noontime temperature measurement with uncertainties less than 1 K up to 1 km for 1 min integration time. Weng et al. (2018) introduced a new PRR lidar system that effectively detects two isolated $N_2$ molecule PRR line signals and elastic backscatter signals. With this new system, temperatures at any given time can be obtained without calibration.

The accuracy of the traditional Raman temperature estimations highly depends on the estimation of the calibration function and calibration coefficients.  Zuev, Vladimir V., et al. has investigated the use of nonlinear calibration functions to improve the accuracy of the traditional Raman temperature estimation and somehow compensate for the heavy assumption of single PRR line. The results showed the $1/T$ term expressed in a form of a quadratic function of log of the ratio of the PRR measurements is the best for practical use.  Another study by He, Jingxi, et al. proposed a new calibration method for PRR lidar temperature profiling based on the different temperature sensitivities of Stokes and anti-Stokes PRR lines. In this paper they reconstruct the expression of

the differential backscatter cross section according to the temperature dependencies of each component and form a temperature factor and a calibration factor in the intensity ratio. This new method has reduced the temperature error by ~50% compared with the commonly-used calibration methods in conditions of low signal-to-noise ratio (SNR).

Temperature profiling from Raman lidar backscatter measurements can also be improved using retrieval schemes based on OEM as we introduce in this paper. A previous study by Yan, Qing, et al. has also proposed an optimized retrieval method for traditional Raman temperature profiling. The proposed method allowed independent alternating solutions to high- and low-quantum-number PRRSs, where high-quantum-number PRR lidar returns are used to solve the channel constant, and low-quantum-number PRR returns with high SNR are used for retrieving temperature profiles. The results showed that the effective temperature retrieval height greatly improved from 17 to 25 km under clear weather conditions and better than 5 K can be obtained up to 25 km.

You mention two assumptions on page 8, line 25/26. What do you mean with "well known"? How critical are these assumptions? When highlighting which assumptions are NOT needed (see abstract, introduction, conclusions), you should be fair and also mention which are needed.

"Well known" was a poor choice of words, we meant the *a priori* extinction profile is based on a backscatter ratio measurement by the lidar, and that the overlap is based on estimates from clear sky measurements compared to the expected overlap (please see our response to your comment on page 15 in the Minor Comments section which explains this in more detail). To estimate an extinction profile requires the assumption of a lidar ratio.

As part of our response to Referee 1 (Christoph Ritter) comments we have added the following paragraph to the paper that explains our approach of the overlap and extinction retrievals.

"The effect of geometrical overlap and particle extinction on the signals are strongly coupled and hence retrieving both parameters simultaneously with the given data channels is not possible unless at least one of the effects is highly constrained. We assume that particle extinction is well known from the backscatter ratio outside clouds, and that overlap is well known above the height of full overlap, i.e. above 6 km (Dinoev et al., 2010).We use this knowledge to define a transition height, 6 km in clear skies or at the cloud base height, whatever is lower. Below this height overlap is retrieved, and above this height particle extinction is retrieved. The *a priori* overlap function is estimated from measurements in clear sky conditions. A 50% standard deviation is used for geometrical overlap below the transition height and a constant standard deviation of $10^{-3}$ is used above this height, constraining the geometrical overlap to the *a priori* values above the transition height. For particle extinction, a standard deviation of $10^{-6}km^{-1}$ is used

below the transition height to constrain the retrieval, then a 50% standard deviation is used above this height, allowing the OEM to retrieve exclusively the particle extinction. The *a priori* covariance matrices for both particle extinction and geometrical overlap are determined using a tent function with a 100m correlation length."

Maybe it would be interesting to explain in the abstract and conclusions that (how much) the results are independent from the selected *a priori* temperature profile.

Similar to Sica and Haefele (2015) , we trust our retrievals upto a certain cutoff height where the response function falls below a value of 0.9. Thus, our retrieval depends on maximum of 10% of the *a priori* temperature profile and that dependency matter only in the heights where the signal strength becomes weaker. One can choose different values of response function to define the cutoff height.
We will add the following sentences to the conclusion explaining the dependency of the temperature retrievals from the selected *a priori* temperature profile.

The OEM retrieved temperatures depend nearly to 100% of the measurement. At the cut off height, the measurement contribution falls below 90% . Temperature retrievals above the cutoff height depend to more than 10% on the *a priori* temperature profile and are not considered an independent measurement.

Page 9, line 4: There are always aerosols in the troposphere. How few are acceptable?

In our study we define the clear and cloudy conditions in the atmosphere based on the backscatter ratio profiles we calculate using RALMO elastic and PRR measurements. When there are no clouds or thick aerosol loads found we have observed the backscatter ratio is less than 2. We consider such cases to be cloud free and retrieve the overlap function up to 6 km.

I think the term "digital measurements" for photon counting signals is odd. Also the analog signals are digitized. Thus, I suggest that you write "photon counting" throughout the text.

We will change the term digital measurements to photon counting.

You are correct, this error was also pointed out by Referee 1. We  made a mistake in the analog measurement units. Our response here is the same as to Referee 1:
"The RALMO analog raw data  are sampled by Licel counters, and then converted to counts (ADC) . However, it does not change the unit of the analog signal. Thus, we have made a mistake in the analog signal units in Figures 1, 7,12 and 18. Units for the analog signals are now corrected to the units of mV."

**Minor comments:**

"Lidar constants" and "coupling constants" are terms which are not commonly used. Please define/explain or avoid. E.g., page 1, line 5: "ratio of efficiencies" is better I think since the laser power, transmitter efficiency, telescope area are the same for both channels and thus cancel and become irrelevant (if I understand correctly).

Lidar constant /lidar system constant  (Liu, Z., Voelger, P., & Sugimoto, N. (2000), Tao, Zongming, *et al*. (2008) ,Winker, D. M., and M. A. Vaughan, Kovalev(1994), Vladimir A., and H. Moosmüller (1994)) refers to combining instrument and physical constants into a single constant. While individual instrument parameters can often not be determined, such as the optical efficiency of the system, the overall value of the constant can be estimated using the return photocount profiles (e.g. Sica *et al.* 1995).  The term coupling constant,  the ratio of the two lidar constants, not a common term. This ratio was called a coupling constant by Sica and Haefele (2016), since it mathematically coupled the lidar equations for physical separate counting channels in their forward model. In this study a coupling constant is used in the forward model to couple the measurements from the two PRR digital/analog channels. We would prefer to define and use the term coupling constant in the manuscript rather than ratio of efficiencies. We will make sure these terms are well defined.

Page 1, line 21: ". . .2 flights per day". One could add: "only at selected sites worldwide".
We will add this.

Page 1, line 22: Please add references to other combined Raman lidar systems. There are several which measure water vapor and temperature.
We have added the following references:

1.  Mattis, Ina, et al. "Relative-humidity profiling in the troposphere with a Raman lidar." *Applied optics* 41.30 (2002): 6451-6462.
2.  Wang, Yufeng, et al. "A detection of atmospheric relative humidity profile by UV Raman lidar." *Journal of Quantitative Spectroscopy and Radiative Transfer* 112.2 (2011): 214-219.
3.  Reichardt, Jens, et al. "RAMSES: German Meteorological Service autonomous Raman lidar for water vapor, temperature, aerosol, and cloud measurements." *Applied optics* 51.34 (2012): 8111-8131.
4.  Behrendt, Andreas, et al. "Combined Raman lidar for the measurement of atmospheric temperature, water vapor, particle extinction coefficient, and particle backscatter coefficient." *Applied optics* 41.36 (2002): 7657-7666.

If we have missed anything else please let us know and we would be glad to add it.

Page 3, Eq. 1: You assume that all PRR lines of one channel are collected with the same efficiency. This is generally not the case but may be true for RALMO. Please add a comment on this.

We do not consider the efficiencies of all the PRR lines are the same. Transmission of the receiver at the wavelength of the PRR line given by $\tau_{RR}(J_i)$ in Eq.1 represents the efficiencies of each detected by the RALMO system. $\tau_{RR}(J_i)$ for RALMO are known (Figure 1, Dinoev, T. S., *et al*) and used in our forward model.

Page 3, Eq. 2: This equation is not found in Penney et al. 1974.

This equation is from Behrendt A (2005) and the original equations are based on Penney et al. 1974. We will fix this in the manuscript.

Page 5, line 11: Why is the background noise B_RR a function of height? I think the "real background" should be height independent. If the baseline is height dependent, detector non-linearity or electronic cross-talk is present, I guess.

In general background noise $B_{RR}$ is a function of height. For some systems this can be independent of height. On Page 9 line 11 we have explained that for RALMO background noise is independent of height.

Page 7, table 2: What do you with "transition height"?

We have fixed this to:

| Geometrical Overlap Function | Estimated using the forward model and measurements | 50% **below and at** transition height |
|---|---|---|
| Particle Extinction | Estimated using measurements | $10^{-6}$km$^{-1}$ **below and at** transition height |

Page 8, line 6: beta_par *is* related

Changed the text.

Page 9, line 20: I would prefer "model parameters" or "model b parameters"

Changed the text.

This sentence is now changed to "The traditional temperature profiles that will be shown later in this section are calculated using count profiles consisting of glued analog and digital measurements which are corrected for non-linearity and background before processing".

Page 10: line 6: What do you mean with "dominant"?
In lower altitudes signal strengths are higher and as the height increases the signal gets weaker. Then the effect from the electrical offset starts to dominate and analog signal in higher altitudes will have values that are not correct.
As we think the word dominant poor choice of words we will change this to be "becomes larger."

Page 10, line 7: How do you know that the analog signals are linear?
RALMO uses Licel transient recorders for data acquisition with Licel's Hammatsu PMTs. The Licel transient recording for analog measurements are designed/tested for linearity by the manufacturer. These systems are widely used in the lidar community, and we are not aware of any evidence that the analog channels are nonlinear.

Page 10, line 8: "become saturated". What do you mean with "saturated"?
What we are implying here is that the photon counting measurements that are above 10MHz are no longer linear.  We have made a mistake in the paper by stating the saturation limit is 2 MHz. We will correct this to 10MHz.

Page 11, Fig 2: I suggest that you write "four signals". It is *two* channels.
Will change the text.

How did you determine "measurement noise"? With OEM? Please clarify.
The OEM does not determine measurement noise. However, we require measurement noise to evaluate the OEM. As given in page 7 line 5, for photon counting measurements which are linear, the measurement variance is equal to the square root of the photon counting measurement (Poisson statistics). For photon counting measurements that are non-linear and for analog measurements we use the auto-covariance method (refer Lenschow *et al.* (2000). ) to estimate the measurement noise.

Page 14, line 2: What is coupling constant R_a? How is it defined?
$R_a$ is the coupling constant for analog channels. It is estimated using Eq.14. This has been stated in the paper as " The coupling constants for analog ($R_a$ ) and digital (R ) channels are estimated by fitting the ratio of PRR measurements with the ratio of the differential cross section (Eq.( 14))."

The *a priori* overlap function used in the OEM scheme is estimated using the OEM retrievals of temperature during clear sky conditions. A few clear day and nighttime measurements were processed in the OEM using a model overlap function given in Dinoev, T., *et al*. (2013)  and corresponding overlap functions were retrieved. Then the average of the retrieved overlap functions from the measurements obtained at clear conditions were used as the *a priori* overlap function in the current PRR temperature OEM scheme.

The temperature retrieval is not sensitive to the choice of the *a priori* overlap function. Both, a sensitivity analysis and the retrieval diagnostics (the overlap averaging kernels) (see below) confirm this. But the overlap function is still important since the lidar signal for each channel is modelled explicitly, as opposed to the traditional method, which works with the signal ratios. Analogously, in the traditional method, the overlap could be determined using the retrieved temperature profile, a forward model and an extinction profile from an ancillary source.

[Figure]

Figure: Left Panel : The OEM retrieved overlap (red curve) and the a priori overlap (green curve) from the clear nighttime RALMO measurements made on 20110909. Right Panel: The overlap averaging kernel.

The reason we are not showing the analog signals in the same height range as digital is that as in altitudes above 15 km analog signals go to background. Thus, the variations in the lower level analog signals are not clearly visible with the extended altitude axis.

We have only showed the elastic signal for Case study 4 as it is a special case. However, we will attach the profiles of elastic signals for the other 3 cases studies as a supplementary document.

Eb is a term that is used in the RALMO system, that refers to the **e**lastic backscatter detected by the **b**ig telescope.

**The language still needs polishing/corrections. Here two examples:**
Page 1, line 3ff: "assumption for the form of . . ." However, I think the form of the calibration function is not really the point here. Calibration with external sensors (with all the uncertainties related to the accuracy of the reference sensor and to the sampling differences) is usually needed but overcome with OEM.

Thank you. We will go over the text again carefully before submitting. We have fixed the 2 unclear sentences you have mentioned above.

The paper describes the temperature retrieval by lidar using the optimal estimation method. Four different cases (each day / night and clear / cloudy) from Payerne station are used to demonstrate the feasibility of the method. While the OEM method for temperature retrieval is not entirely new and the first and third bullet point of the conclusions may be found elsewhere, the paper as a whole contents enough information for publishing. It is generally well written and well structured. The 4 selected cases are new and demonstrate the usage of the still not too far spread OEM technique.

I have a short list of questions and remarks to the author which should be clarified prior to publishing:

You show a complete error analysis for the OEM technique, which I found convincing. However, to better judge these results it would be very interesting to have a similar error estimation for the traditional Raman technique. Especially in your case 1 fig. 4 it seems as if the Raman result is only plotted up to 12 km although in the manuscript a change of vertical resolution in this altitude is mentioned (page 11). Can you comment on this?

Do you assume the same overlap for both channels? Why do you need the overlap? Do you really think to be able to retrieve the overlap with the required precision to obtain aerosol extinction information? In this case an error analysis would be required, otherwise revise your wording. I think for this paper such an effort is not necessary – as the particle extinction is the same for the high and low channel, hence the temperature from the OEM should not depend on the extinction?

Page 11 and Fig. 5: your error analysis is nice and one of the strong selling points of this paper. However, in the current form I cannot reproduce the values. A bit more information is required, how the values were obtained.

Minor points:

Page 1 quote Mahagammulla Gamage: I think in the introduction it is not necessary to quote a paper which is still under preparation. You may choose another quote here

Page 3: (and elsewhere) is d sigma / d Omega really the ATTENUATED cross section? I am not sure, as you have Gamma^2 as extinction term in your eq. 1.

Page 4: explain $x_a$ in eq. 7.

Page 5: minus sign in eq. 8 is missing

Page 9: 2 times "from" in line 14

Page 10, fig. 1: is the units of your analog signal MHz, how did you convert it?

Page 11: line 13: reduced

Page 13: line 4: agrees, line 4: deviates

Page 19: a lidar ratio of 5sr is already quite small, can you estimate an error for the lidar ratio?

Page 20: line 8: "by" missing … that by the OEM method; line 9: I don't understand the "temperature range" – I thought the OEM only depends on K and K_a?

Page 21: line 11: what do you mean be "uncorrected" PRR measurements?

AMT-2019-107: Retrieval of Temperature From a Multiple Channel Pure Rotational
Raman-Scatter Lidar Using an Optimal Estimation Method (Gamage et al.)
Response to Referee 1 (**Christoph Ritter** )
04 July 2019

The Referee's comments are in blue, responses are in black, and the texts added to the manuscripts are in red.

- You show a complete error analysis for the OEM technique which I found convincing. However to better judge these results it would be very interesting to have a similar error estimation for the traditional Raman technique. Especially in your case 1 fig4 it seems as if the Raman result is only plotted up to 12 km although in the manuscript a change of vertical resolution in this altitude is Mentioned page11. Can you comment on this?

We strongly agree that the community of users of the traditional temperature method would benefit from a detailed investigation of that technique in the spirit of the 3 papers by the group of NDACC lidar scientists who have done comprehensive studies of the traditional ozone and Rayleigh scatter temperature determinations, as well as quantifying the vertical resolution of the determinations [see the references at the end of this reply], and we would be glad to participate in that effort. It is, however, tangential to the purpose of this manuscript which is to show a new way to retrieve temperature, which offers some advantages (and disadvantages) relative to the traditional method. For instance in Jalali et al. 2018 and Farhani et al. 2019 the OEM uncertainty budgets were shown to compare well with previous studies for Rayleigh scatter temperature profiles and DIAL ozone profiles (including the NDACC work). These comparisons give us confidence in our uncertainty budgets for lidar retrievals using OEM.

As far as to our choice of plotting the traditional method up to 19km (Fig4: traditional temperatures goes up to 19km not 12km ), the traditional temperature estimates are derived from the MeteoSwiss routine temperature analysis, which works as follows.

The vertical resolution is not constant; it starts at 30m and it increases to 400m.
- The change in vertical resolution is based on the calculated uncertainty of the temperature profile at each height. The maximum height resolution allowed is 400m, until the temperature statistical uncertainty becomes smaller than a threshold value of 1K.
- The calculated profile is cut off based again on the uncertainty. When the algorithm can't calculate the next range gate within the threshold value of 1K it stops the profile at the last range gate.

In Fig. 4 the traditional retrievals are shown up to ~19km. For that specific case (Case 1), the traditional temperature vertical resolution is 30m up to 12.5km, then changes to 400 m above this height. The temperatures are cutoff at the height where the uncertainty reaches 1K (19km).

Above information on the traditional method is given in Section 5 of the manuscript.

- Do you assume the same overlap for both channels? Why do you need the overlap? Do you really think to be able to retrieve the overlap with the required precision to obtain aerosol extinction information? In this case an error analysis would be required otherwise revise your wording. I think for this paper such an effort is not necessary – as the particle extinction is the same for the high and low channel, hence the temperature from the OEM should not depend on the extinction?
  Yes, we assume the same overlap for both channels based on the work of Dinoev et al. 2012. We need the overlap since in OEM we forward model the raw backscatter profile of each channel and hence, have to specify an overlap function. At a given altitude, it is not possible in our retrieval to determine both overlap and extinction. Therefore, our approach is to retrieve whatever quantity we know less well. Generally this is overlap below 6km and extinction above 6km, since above 6km full overlap can be safely assumed. In case of clouds with a ceiling below 6km, we retrieve overlap up to the ceiling and extinction above. Furthermore, the results for extinction and lidar ratio look reasonable compared to other studies. One benefit of OEM is the ability to estimate the effect of a model parameter on the retrieved quantity, so given we have made reasonable choices for the model parameter uncertainties their impact on the retrieved temperature is well characterized.

Moreover, we trust the extinction retrieval and its uncertainty above 6km but not below and we changed the text in Section 4.3 of the manuscript as shown below to clarify this.

"The effect of geometrical overlap and particle extinction on the signals are strongly coupled and hence retrieving both parameters simultaneously with the given data channels is not possible unless at least one of the effects is highly constrained. We assume that particle extinction is well known from the backscatter ratio outside clouds, and that overlap is well known above the height of full overlap, i.e. above 6 km (Dinoev et al., 2010).We use this knowledge to define a transition height, 6 km in clear skies or at the cloud base height, whatever is lower. Below this height overlap is retrieved, and above this height particle extinction is retrieved. The a priori overlap function is estimated from measurements in clear sky conditions. A 50% standard

deviation is used for geometrical overlap  below the transition height and a constant standard deviation of $10^{-3}$ is used above this height, constraining the geometrical overlap to the a priori values above the transition height. For particle extinction, a standard deviation of $10^{-6}km^{-1}$ is used below the transition height to constrain the retrieval, then a 50% standard deviation is used above this height, allowing the OEM to retrieve exclusively the particle extinction. The a priori covariance matrices for both particle extinction and geometrical overlap are determined using a tent function with a 100m correlation length."

Figure 1, shows the full uncertainty budgets of the overlap and particle extinction retrievals for the Case 3: Nighttime with cirrus cloud given in the manuscript. Overlap uncertainties are shown up to 6km and the particle extinction uncertainties are shown above 6 km.

[Figure]

Figure 1: Left Panel: Full uncertainty budget of the overlap retrievals from the nighttime RALMO measurements on 05 July 2011 with a cirrus cloud present at 6km height. Uncertainty due to Rayleigh cross section, pressure, and coupling constants are in the orders of $10^{-4}$, $10^{-1}$, and $10^{-1}$ respectively. Right Panel: Full uncertainty budget of the particle extinction. Uncertainty due to Rayleigh cross section $10^{-3}$ and due to analog coupling constant is $10^{-2}$. Uncertainty due to pressure and digital coupling constants are about 1% and 2% respectively.

- Page 11 and Fig . 5 your error analysis is nice and one of the strong selling points of this paper. However, in the current form I cannot reproduce the values . A bit more information is required, how the values were obtained.

Forward model uncertainties are calculated based on the theory presented in the Rodgers textbook Section 3.2.2. We have also added the following equations that we used to compute the errors into the revised manuscript.

The uncertainty budget is determined from the measurement and model parameter covariance matrices (Rodgers, 2000). The total covariance $S_{total}$ is:

$S_{total} = S_m + S_F$

where $S_m$ is the retrieval covariance due to measurement noise and $S_F$ is the retrieval covariance due to the forward model parameter uncertainty. The retrieval covariance due to measurement noise $S_m$ is

$S_m = GS_yG^T$

where G is the gain matrix that indicates the sensitivity of the retrieval to the measurements. The retrieval covariance due to the forward model parameters $S_F$ is

$S_F = GK_bS_bK^T_bG^T$

where $K_b$ and $S_b$ are the forward model parameter Jacobian and covariance matrices respectively. The model parameter Jacobians $K_b$, can be estimated analytically or numerically for each model parameter. To construct $S_b$ we require the uncertainties of the model parameters. We recommend Rodger's textbook (Rodgers, 2000) for more details of the OEM.

Furthermore, in our work presented in the manuscript we have used the uncertainties used in the Table 2 in the manuscript to construct construct $S_b$ .

- Page 1 quote Mahagammulla Gamage : I think in the introduction it is not necessary to quote a paper which is still under preparation. You may choose another quote here :
  agree will remove this.

- Page 3 : and elsewhere is d sigma /d Omega really the ATTENUATED cross section? I am not sure,  as you have Gamma^2 as extinction term in your eq. 1 ?
Yes. this term is attenuated cross section. We define attenuated differential cross section term as the convolution of the  instrument function with each line in the spectrum.  This is shown in EQ 2, where attenuated cross section contains the terms  $\tau^+ (J_i)$ and  $\tau^- (J_i)$ that are the transmission of the receiver at the wavelength of each RR line. However, this term is not a standard one and it is easy to see why it could be confused with attenuation due to atmospheric transmission. The atmospheric transmission is defined in EQ 1 in the usual manner.

- Page 10..fig. 1 : is the units of your analog signal MHz.how did you convert it? :

However, in the OEM it doesn't matter what unit we use for signal as long as the forward model generated signal and the real measurements are in the same unit. Another advantage of OEM is that multiple detector channels, both analog and digital with differing vertical resolutions and units can be easily used. Thus, these changes have no effect on the results in the manuscript.

- Page19:a lidar ratio of 5sr is already quite small. Can you estimate an error for the lidar ratio?
We can estimate a statistical uncertainty using standard error propagation. The lidar ratio is not directly retrieved in our OEM. Our estimate of LR is based on the retrieved particle extinction and ASR profiles calculated from elastic and PRR lidar measurements. Even though it is small mathematically it is possible to estimate an error and that is what we have done. We will add a phrase to the manuscript explaining how the uncertainty of lidar ratio was calculated. We have also updated Figure 17 in the manuscript, indicating the estimated LR errors.

- Page20: line9: I don't understand the "temperature range"– I thought the OEM only depends on K and K_a?
The calibration coefficients of the calibration function used in the traditional temperature retrievals needs to be estimated over a wide range of possible temperatures. However, our OEM calibration constants (coupling constants) can be estimated over a narrow range of temperatures, or even one single point, without introducing extrapolation errors in the retrieval. The OEM depends on the two coupling constants $R$ and $R_a$ and estimation of those require temperatures, but does not require the assumption of a functional form relating these constants to temperature (refer to Eq.10 and 11 in the manuscript); the requirement of a functional form depending on multiple parameters rather than a constant is why the traditional method must use a wide range of temperatures.

- Page21:line11:what do you mean be"uncorrected "PRR measurements? :
This was meant to be raw PRR measurements, that is a Level 0 product not corrected for saturation or background. We will change this word from "uncorrected" to "raw".

Thank you for noticing the following typos. We will fix these mistakes in the revised paper

- Page 4 : explain x_a in eq. 7. : x_a is the a priori.
- Page 5 : minus sign in eq. 8 is missing : Fixed.
- Page 9 : 2 times "from" in line 14 : Fixed
- Page 11 : line 13 : reduced : Fixed
- Page13: line 4: agrees line 4:deviates: Fixed
- Page20:line8:"by"missing…that by the OEM method[a] : Fixed

35 are used to solve for the channel constant, while low quantum number PRR returns with higher SNR are used for retrieving

[revised manuscript text omitted]
 $\boldsymbol{y}$ to estimate the state (retrieval) variables $\boldsymbol{x}$ of a system via a forward model. The forward model $\boldsymbol{F}$ contains all the atmospheric and instrumental physics that describe the measurements. The

15      forward model can include model parameters $\boldsymbol{b}$, which are assumed and not retrieved, and their effect on the retrieved quantity uncertainties can be calculated.

     The measurements are related to the forward model by:

$$\boldsymbol{y} = \boldsymbol{F}(\boldsymbol{x}, \boldsymbol{b}) + \epsilon \tag{6}$$

where $\epsilon$ represents measurement noise. Under the assumption that all parameters have Gaussian probability density functions

20      Bayes' theorem can be applied to determine the cost function,

$$cost = [\boldsymbol{y} - \boldsymbol{F}(\boldsymbol{x}, \boldsymbol{b})]^T \boldsymbol{S}_y^{-1} [\boldsymbol{y} - \boldsymbol{F}(\boldsymbol{x}, \boldsymbol{b})] + [\boldsymbol{x} - \boldsymbol{x}_a]^T \boldsymbol{S}_a^{-1} [\boldsymbol{x} - \boldsymbol{x}_a], \tag{7}$$

where $\boldsymbol{x_a}$ is the *a priori* of the retrieval parameters, $\boldsymbol{S_y}$ is the measurement covariance, which describes the random measurement uncertainty and $\boldsymbol{S_a}$ is the *a priori* covariance. The cost function measures the goodness of fit for a solution, and for good models the cost is on the order of unity. For non-linear forward models, the Marquardt-Levenberg method can be used

25      iteratively to minimize the cost of the retrieval (see Section 5.7 in Rodgers (2000) for details).

     The uncertainty budget is determined from the measurement and model parameter covariance matrices (Rodgers, 2000). The total error covariance, $\boldsymbol{S_{total}}$, is:

$$\boldsymbol{S_{total}} = \boldsymbol{S_m} + \boldsymbol{S_F} \tag{8}$$

where $S_m$ is the retrieval covariance due to measurement noise and $S_F$ is the retrieval covariance due to the forward model parameter uncertainty. The retrieval covariance due to measurement noise, $S_m$, is

$$S_m = GS_yG^T \qquad (9)$$

where $G$ is the gain matrix which is the sensitivity of the retrieval to the measurements. The retrieval covariance due to the forward model parameters, $S_F$, is

$$S_F = GK_bS_bK_b^TG^T, \qquad (10)$$

where $K_b$ and $S_b$ are the forward model parameter Jacobian and covariance matrices, respectively. The model parameter Jacobians $K_b$, can be estimated analytically or numerically for each model parameter. To construct $S_b$ we require the uncertainties of the model parameters. We recommend Rodgers (2000) for more details of the OEM.

**4.2 The forward model for a PRR lidar**

The forward model describes the measurement as a function of both the state of the atmosphere and instrumental parameters. The core of our forward model is the lidar equation as presented in Section 3. It is called 4 times to generate the measurements corresponding to high and low quantum numbers, i.e. JH and JL, with digital and analog detection. Analog detection is assumed to be linear and hence the saturation equation (Equation 5) is not applied.

The pressure, $P(z)$, and temperature, $T(z)$, can be taken from either a radiosonde measurement or an atmospheric model. The background noise, $B_{RR}$, is in general a function of height, $z$, but is constant with height for RALMO. Unlike all the other existing forward models for lidar except Povey et al. (2012) (which was designed specifically to determine overlap) we retrieve $O(z)$ the geometrical overlap function in addition to temperature. The overlap functions of JH and JL channels were assumed to be the same (Dinoev et al., 2010).

The atmospheric transmission, $\Gamma_{atm}(z)$ in Eq.( 1), includes both molecular and particle scattering.

$$\Gamma_{atm}(z) = \exp\left[-\int_0^z (\alpha_{mol} + \alpha_{par})dz\right] \qquad (11)$$

where $\alpha_{mol}$ is the molecular extinction coefficient and $\alpha_{par}$ is the particle extinction coefficient. The molecular extinction can be expressed using the Rayleigh cross section $\sigma_{Ray}$ and air density $n_{air}$ as:

$$\alpha_{mol} = \sigma_{Ray}.n_{air} \qquad (12)$$

where $\sigma_{Ray}$ is calculated using the expressions given by Nicolet (1984).

For each channel the subscript $RR$ is replaced by JL and JH ,the high and low quantum number PRR channels. Then $C_{RR}$, $B_{RR}$ and $J_i$ then have different values.

RALMO detects multiple Stokes and anti-Stokes lines from both nitrogen and oxygen PRR spectrum. Therefore, to determine the attenuated backscatter cross-section in the forward model we require knowledge of the exact number of quantum

states detected by each the RALMO PRR channel. From the JH and JL channel characteristics we can calculate the range of frequency shifts for each channel relative to the elastic wavelength 354.7 nm. Then using the equations given by Herzberg (2013) we can determine the quantum numbers $J_i$ for both nitrogen and oxygen molecules. This calculation process is repeated to determine the $J_i$ numbers for the JL channel of the RALMO. The summary of the findings are given in Table 1.

**Table 1. The respective quantum lines from nitrogen and oxygen PRR spectrum detected by the RALMO temperature polychromator. Note the given quantum lines are valid for both the Stokes and anti-Stokes branches.**

| Channel | | Nitrogen | Oxygen |
|---------|--------------|-----------------|-------------|
| JL | Quantum lines | 3,4,5,7,8 | 5,7,9,11 |
| JH | Quantum lines | 10,11,12,13,14,15 | 15,17,19,21 |

[revised manuscript text omitted]
 (Dinoev et al., 2010). We use this knowledge to define a transition height, in clear skies 6 km or at the cloud base height, whatever is lower. Below this height overlap is retrieved, and above this height particle extinction is retrieved. The *a priori* overlap function is estimated from measurements in clear sky conditions. A 50% standard deviation is used for geometrical overlap below the transition height and a constant standard deviation of $10^{-3}$ is used above this height, constraining the

geometrical overlap to the *a priori* values above the transition height. For particle extinction, a standard deviation of $10^{-6} \text{km}^{-1}$ is used below the transition height to constrain the retrieval, then 
[revised manuscript text omitted]

Weng, M., Yi, F., Liu, F., Zhang, Y., and Pan, X.: Single-line-extracted pure rotational Raman lidar to measure atmospheric temperature and aerosol profiles, Opt. Express, 26, 27 555–27 571, https://doi.org/10.1364/OE.26.027555, http://www.opticsexpress.org/abstract.cfm?URI=oe-26-21-27555, 2018.

Whiteman, D. N.: Examination of the traditional Raman lidar technique. I. Evaluating the temperature-dependent lidar equations, Applied Optics, 42, 2571–2592, 2003.

Yan, Q., Wang, Y., Gao, T., Gao, F., Di, H., Song, Y., and Hua, D.: Optimized retrieval method for atmospheric temperature profiling based on rotational Raman lidar, Applied Optics, 58, 5170–5178, 2019.

Zuev, V. V., Gerasimov, V. V., Pravdin, V. L., Pavlinskiy, A. V., and Nakhtigalova, D. P.: Tropospheric temperature measurements with the pure rotational Raman lidar technique using nonlinear calibration functions, Atmospheric Measurement Techniques, 10, 315–332, 2017.